# Densest Subhypergraph: Negative Supermodular Functions and Strongly Localized Methods

## ABSTRACT

Dense subgraph discovery is a fundamental primitive in graph and hypergraph analysis which among other applications has been used for real-time story detection on social media and improving access to data stores of social networking systems. We present several contributions for localized densest subgraph discovery, which seeks dense subgraphs located nearby a given seed sets of nodes. We first introduce a generalization of a recent *anchored densest subgraph* problem, extending this previous objective to hypergraphs and also adding a tunable locality parameter that controls the extent to which the output set overlaps with seed nodes. Our primary technical contribution is to prove when it is possible to obtain a strongly-local algorithm for solving this problem, meaning that the runtime depends only on the size of the input set. We provide a strongly-local algorithm that applies whenever the locality parameter is at least 1, and show why via counterexample that strongly-local algorithms are impossible below this threshold. Along the way to proving our results for localized densest subgraph discovery, we also provide several advances in solving global dense subgraph discovery objectives. This includes the first strongly polynomial time algorithm for the densest supermodular set problem and a flow-based exact algorithm for a densest subgraph discovery problem in graphs with arbitrary node weights. We demonstrate the utility of our algorithms on several web-based data analysis tasks.

## CCS CONCEPTS

• **Theory of computation → Network flows**; **Network optimization**; *Linear programming*; • **Mathematics of computing → Hypergraphs**; • **Information systems** → *Web mining*.

## KEYWORDS

Densest Subgraph, Strongly Localized Graph Algorithms, Hypergraph Algorithms, Maximum Flow, Supermodular Functions

**ACM Reference Format:**
Anonymous Author(s). 2018. Densest Subhypergraph: Negative Supermodular Functions and Strongly Localized Methods. In *Proceedings of Make sure to enter the correct conference title from your rights confirmation emai (Conference acronym 'XX).* ACM, New York, NY, USA, 14 pages. https://doi.org/XXXXXXX.XXXXXXX

*Conference acronym 'XX, June 03–05, 2018, Woodstock, NY*
© 2018 Association for Computing Machinery.
ACM ISBN 978-1-4503-XXXX-X/18/06...$15.00
https://doi.org/XXXXXXX.XXXXXXX

## 1 INTRODUCTION

A common paradigm in unsupervised data analysis is to take as input a graph or hypergraph and to output extremal subsets. The types of extremal subsets range from sets of minimum cut [39] to minimum sparsest cut [33] to maximal clique [8]. The underlying hypothesis is that extremal sets reflect important and noteworthy structures that are informative for exploratory data analysis, or useful for downstream algorithms such as graph partitioners or machine learning pipelines that operate on subsets of the larger graph. This basic paradigm is fundamental in Web analysis and applied to problems such as detecting real-time stories on social media [6], improving access to data stores of social-networking systems [21], and a wide variety of clustering and community detection tasks over web-based datasets [4, 18, 30, 45].

An issue with this paradigm is that there are many cases where extremal sets are trivial or simply unuseful. For instance, the minimum cut set in an unweighted graph with any degree 1 node is just that single node, which yields little information; a set of minimum conductance may be a simple small subgraph that just happens to have a small number of bridges to the rest of the graph [30]. A second and related challenge is that finding the extremal subset typically results in an NP-complete problem. Even if solved approximately, it may still consume substantial time.

*Localized* graph algorithms are a practical solution to this problem. The idea is that we rephrase the extremal search problem with respect to a reference set of nodes $R$. For instance, we may want the solution *within $R$* or *nearby $R$* with some measure of distance or fraction of $R$. This area has been most developed in the space of algorithms for finding small conductance cuts in a graph where techniques range between spectral methods [3, 4], flow methods [5, 19, 29, 36, 46, 47], and heat-kernel diffusions [27]. These techniques have also been extended to hypergraph analysis [26, 32, 43]. For methods that are able to effectively *grow* small subsets, then $R$ may be as small as a single node; whereas for other techniques that *shrink or adapt $R$*, then $R$ must be considerably larger. Often, a goal with these algorithms is to get a strongly localized runtime guarantee such that the total runtime scales with the size of the output instead of the size of the input graph. Using a localized algorithm enables one to analyze *many* interesting sets in the graph by varying the reference set $R$. These localized algorithms have already been widely used in web-based data analysis tasks such as detecting related retail products on Amazon [27, 42, 46], identifying groups of same-topic posts on Stackoverflow [42], clustering restaurants based on reviews on Yelp [32], and finding communities in various types of online social networks [27, 46, 47].

Although many extremal set problems in graph analysis focus on finding small *cut* values, another perspective on extremal sets seeks *high density* independently of cut values. The *densest subgraph* is one such example that seeks a subgraph $S$ of maximum average degree. In a small surprise, this subset can be computed in polynomial

time by a classic flow algorithm [22] or via linear programming [10]. A simple peeling algorithm that removes vertices from the graph one at a time has long been known to be a 2-approximation for the problem [10]. More recently, an iterated peeling algorithm has been shown to converge to the optimal solution [11]. Many variants and generalizations of the densest subgraph problem have been studied and considered (see [28] for a very recent survey). One of the most general of these is the densest supermodular subset problem (DSS), where the goal is to maximize the ratio between a nonnegative monotone supermodular function $f$ and the size of the returned set. Localized variants of the densest subgraph problem have also been considered recently [15]. However, localized algorithms for dense subgraph discovery remain underexplored and remain far less understood than localized algorithms for finding small cuts.

In this paper we greatly expand the scope of possible algorithms for dense subgraph computations, both in terms of global and local variants of the problem. We first provide a simple reduction that leads to efficient exact algorithms for a more general version of the densest supermodular subset (DSS) problem where the supermodular function does not need to be nonnegative (Theorem 1). This captures several dense subgraph problems that are not special cases of the standard nonnegative DSS problem [15, 34]. We then provide the first strongly polynomial algorithm for DSS (Algorithm 1, Theorem 2); previous approaches came with weakly polynomial runtimes. Our final contribution to *global* dense subgraph discovery algorithms is to design a flow-based exact algorithm for finding the densest subset of a node-weighted graph or hypergraph. Prior research on this problem showed how to obtain efficient flow-based solutions in the case of graphs with strictly non-negative weights [17, 22]; our results show how this can be extended to arbitrary node weights (Section 5.1).

In addition to our results for global dense subgraph discovery, we greatly advance the state of the art in localized densest subgraph computations (Section 5.2). First, we establish a parametric formulation of the discrete objective function underlying localized densest subgraph discovery (Problems 5, 6). This allows us to vary the degree of localization and continuously tradeoff between the degree of localization and the amount of computation. We explicitly delineate the region of strong locality where algorithms can have a runtime that scales independently of graph size (Theorem 4). Moreover, we show hypergraph generalizations of all of these algorithms. Our methods use max-flow / min-cut computations as a primitive and we show (in the appendix) counter-examples where standard *peeling* methods cannot approximate these objectives at all.

We demonstrate the advantages of the techniques on a variety of web-relevant datasets. This includes a hypergraph of web domains where hypergraphs are induced by hosts (Section 6.1). We show how our localized algorithms can help identify a densely connected set of about 1300 academic domains around the world.

## 2 PRELIMINARIES AND RELATED WORK

Let $G = (V, E)$ denote a graph with vertex set $V$ and edge set $E$. Let $\mathcal{H} = (V, \mathcal{E})$ be a hypergraph with vertex set $V$ and hyperedge set $\mathcal{E}$. Each hyperedge $e \in \mathcal{E}$ is a subset of $V$ and a graph is the special case of a hypergraph with $|e| \leqslant 2$. Our results for hypergraphs focus on unweighted and undirected hyperedges without self-loops, though

**Table 1: Two kinds of degrees we consider.**

| | Normal | Fractional |
|---|---|---|
| Degree | $\deg(v) = \sum_{e \ni v} 1$ | $\overline{\deg}(v) = \sum_{e \ni v} \frac{1}{|e|}$ |
| Volume | $\mathrm{Vol}(S) = \sum_{v \in S} \deg(v)$ | $\overline{\mathrm{Vol}}(S) = \sum_{v \in S} \overline{\deg}(v)$ |
| Maximum | $\Delta(S) = \max_{v \in S} \deg(v)$ | $\bar{\Delta}(S) = \max_{v \in S} \overline{\deg}(v)$ |

the techniques can be easily extended to weighted hyperedges. At the same time, our results in some cases rely on reductions to weighted and directed graphs. By default, we use $uv$ to denote a directed edge from vertex $u$ to $v$, and a set of nodes $\{v_1, \ldots, v_k\}$ to denote a hyperedge. For a hypergraph, let $r = \max_{e \in \mathcal{E}} |e|$ denote its rank. For a (hyper)graph and a set $S$, let $e[S]$ denote the number of (hyper)edges fully contained in $S$. For any vector $q \in R^V$ and vertex set $S \subset V$, let $q(S) = \sum_{v \in S} q(v)$ denote the summation of entries of $q$ indexed by $S$.

Throughout this paper, we mainly consider two kinds of degrees, the definitions of them and their corresponding volumes and maximums are summarized in Table 1. We have the following connection between degree and fractional degree.

LEMMA 1. *For a hypergraph, we have*

$$2\overline{\deg}(v) \leqslant \deg(v) \leqslant r\overline{\deg}(v), \quad \begin{array}{l}\text{and as} \\ \text{a result}\end{array} \quad 2\overline{\mathrm{Vol}}(S) \leqslant \mathrm{Vol}(S) \leqslant r\overline{\mathrm{Vol}}(S).$$

PROOF. Because we focus on hypergraphs without self-loops, $2\overline{\deg}(v) = \sum_{e \in \mathcal{E}: e \ni v} \frac{2}{|e|} \leqslant \sum_{e \in \mathcal{E}: e \ni v} 1$. Since $r = \max_{e \in \mathcal{E}} |e|$, $r\overline{\deg}(v) = \sum_{e \in \mathcal{E}: e \ni v} \frac{r}{|e|} \geqslant \sum_{e \in \mathcal{E}: e \ni v} 1$. □

To evaluate any set function $f : 2^V \to \mathbb{R}$ we assume it is available as a value oracle as is standard practice. A set function $f$ is normalized if $f(\emptyset) = 0$ and nonnegative if $f(S) \geqslant 0, \forall S \subseteq V$. Further, let $EO(f)$ be the maximum amount of time to evaluate $f(S)$ for a subset $S \subseteq V$ and $M(f)$ be an upper bound for $|f(S)|$ for all $S \subseteq V$. A set function $f$ is supermodular if and only if $f(S) + f(T) \leqslant f(S \cap T) + f(S \cup T)$ for any $S, T \subseteq V$, and accordingly $f$ is submodular if $-f$ is supermodular. A function is modular if it is both supermodular and submodular. Note that a normalized, nonnegative and supermodular set function $f$ is monotone.

### 2.1 Graph Cut and Hypergraph Cut

Densest subgraph discovery(DSG) has a close connection with graph cut problems as the decision version of DSG is solvable by reducing it to a graph min $s$-$t$ cut problem [22]. Here we briefly introduce some graph cut concepts that will show up in the following discussion. For a weighted directed graph $G = (V, E, w : E \to \mathbb{R})$ and a set $S$, the value of its induced cut is $\mathbf{cut}_G(S) = \sum_{u \in S, v \in \bar{S}} w_{uv}$. The graph min $s$-$t$ cut problem is to find the minimal graph cut while enforcing $s \in S$ and $t \in \bar{S}$. In other words, $\mathbf{min\text{-}st\text{-}cut}_G = \min_{S \subset V: s \in S, t \in \bar{S}} \mathbf{cut}_G(S)$.

The introduction of hyperedges enables a variety of definitions of cut as one hyperedge can be cut in more than one way now. Here we adopt a recent generalized notion of a hypergraph cut function [31, 42]. Given a hypergraph $\mathcal{H} = (V, \mathcal{E})$, associate each hyperedge $e$ with a splitting function $w_e : 2^e \to \mathbb{R}_{\geqslant 0}$ that maps each subset $A \subseteq e$ to a nonnegative splitting penalty. The value

$w_e(A)$ indicates the penalty when $S \cap e = A$. Then for one vertex set $S \subseteq V$, the cut penalty it incurs is $\mathbf{cut}_{\mathcal{H}}(S) = \sum_{e \in \mathcal{E}} w_e(S \cap e)$. The corresponding hypergraph min $s$-$t$ cut problem is $\mathbf{min\text{-}st\text{-}cut}_{\mathcal{H}} = \min_{S \subset V: s \in S, t \in \bar{S}} \mathbf{cut}_{\mathcal{H}}(S)$.

## 2.2 Related Work

The classic densest subgraph problem is defined as

**PROBLEM 1 (DENSEST SUBGRAPH (DSG)).** *Given a graph $G = (V, E)$, find a vertex set $S$ maximizing the fraction $e[S]/|S|$.* [1]

DSG and its variants have received significant attention over the past a few decades. They mainly admit two categories of exact solutions, one is flow-based [22] and the other one is based on a linear program (LP) [10]. One popular approximation algorithm for DSG and some variants is greedy peeling [10], which runs in linear time and is much faster than exact solutions. One variant of DSG, called densest hypersubgraph (DHSG), is same as Problem 1 except the graph $G$ is replaced by a hypergraph $\mathcal{H}$ [25]. For a detailed introduction, refer to the recent tutorial [41] and survey [28].

Recently [9] introduces an iterative peeling method for Problem 1 called Greedy++ which shows quick convergence to the optimum. Then [11] showed that Greedy++ achieves a $(1 - \varepsilon)$-approximation in $O(1/\varepsilon^2)$ iterations and extends iterative peeling to a broader class of problems called densest supermodular subset (DSS).

**PROBLEM 2 (DENSEST SUPERMODULAR SUBSET (DSS) [11]).** *Given a normalized, nonnegative monotone supermodular function $f : 2^V \to \mathbb{R}_{\geqslant 0}$, maximize $f(S)/|S|$.*

This is an important breakthrough because numerous DSG variants are special cases of Problem 2 [17, 22, 25, 40, 44]. Therefore iterative peeling offers a faster algorithm for them compared with flow and LP. Moreover, [24] proposes an even faster and more scalable iterative algorithm for Problem 2 based on solving the quadratic relaxation of the dual of Charikar's LP. Although iterative peeling converges fast in practice, it is hard to terminate as soon as some user-defined approximation ratio is achieved as the optimum is not known in advance. Recently, [17] tackles this issue for a subclass of Problem 2, DSG with nonnegative vertex weights.

Besides the line of designing faster *global* algorithms for DSG and its variants, there is some recent interest in studying seeded variants of DSG where a seed set $R$ is given and the objective is to find a densest subgraph around this seed set [15, 17, 38]. Of these, only [15] provides an objective that gives a strongly local algorithm, meaning that the optimal answer is found only by exploring a small portion of the whole graph, via the objective:

**PROBLEM 3 (ANCHORED DENSEST SUBGRAPH (ADS)).** *Given a graph $G = (V, E)$ and a seed set $R \subset V$, find a vertex set $S$ maximizing $\left(2e[S] - \text{Vol}(S \cap \bar{R})\right)/|S|$.*

In Problem 3, the bias towards the seed set is encoded by adding penalties onto vertices outside the seed set.

## 3 GENERAL DENSE SUPERMODULAR SUBSET

The function $2e[S] - \text{Vol}(S \cap \bar{R})$ in Problem 3 is a normalized supermodular function as $e[S]$ is supermodular and $\text{Vol}(S \cap \bar{R})$ is

---

[1] We always treat $0/0 = -\infty$.

---

modular. But it is not a special case of Problem 2 as this function is not guaranteed to be nonnegative. This inspires our broader class:

**PROBLEM 4 (DENSEST SUPERMODULAR SUBSET WITH POSSIBLE NEGATIVE VALUES).** *Given a normalized supermodular function $f : 2^V \to \mathbb{R}$, maximize $f(S)/|S|$.*

In addition to the anchored densest subgraph objective mentioned above, this new formulation also generalizes the objective $\max_{S \subset V} \left(e[S] - \alpha e(S, \bar{S})\right)/|S|$ considered in [34] . We prove the following connection between this extension and the class DSS.

**THEOREM 1.** *For any normalized supermodular function $f : 2^V \to \mathbb{R}$, one can construct a normalized, nonnegative monotone supermodular function $g : 2^V \to \mathbb{R}_{\geqslant 0}$ such that*

$$\operatorname*{argmax}_{S \subset V} f(S)/|S| = \operatorname*{argmax}_{S \subset V} g(S)/|S|$$

*and the difference between functions $f$ and $g$ can be computed in $O(|V|EO(f))$ time.*

**PROOF.** Let $C = \max\{0, \max_{v \in V} -f(\{v\})\}$, in other words $C$ is the smallest nonnegative quantity such that $C + f(\{v\}) \geqslant 0, \forall v \in V$. Then we construct $g$ as $g(S) := f(S) + C|S|, \forall S \subset V$. Since $C|S|$ is modular, $g$ is still supermodular. Observe that because $f$ is supermodular, for any set $S = \{v_1, v_2, \ldots, v_{|S|}\} \subset V$, we have $g(S) = f(S) + C|S| \geqslant f(S \setminus \{v_1\}) + f(\{v_1\}) + C|S| \geqslant \ldots \geqslant \sum_{i=1}^{|S|} f(\{v_i\}) + C|S| \geqslant 0$ which means that $g(S)$ is nonnegative and implies that $g(S)$ is monotone. Thus,

$$\max_{S \subset V} f(S)/|S| + C = \max_{S \subset V} g(S)/|S|.$$

And $C$ can be computed by querying $f(\{v\})$ for each $v \in V$, which can be done in $O(|V|EO(f))$ time. □

This theorem implies that any exact algorithm for DSS will remain as an exact algorithm for the extended Problem 4, such as using linear programming or combining binary search with repeated submodular minimization.

While extending the definition to non-negative valued functions may seem a minor change as it is easy to adapt exact algorithms, this change has large implications for approximation algorithms. For instance, the efficient greedy peeling fails to hold a constant approximation ratio. In the Appendix B, we show one example where greedy peeling may perform arbitrarily badly. Even for the recent iterative peeling approach [11], the picture is more complex and the bounds are not straightforward. That said, we hypothesize that iterative peeling remains an effective practical heuristic.

## 4 A STRONGLY POLYNOMIAL ALGORITHM

Strongly polynomial algorithms are those having a running time bounded by a polynomial of the number of input numbers instead of their size. In the context of Problem 4, a strongly polynomial algorithm is one whose runtime is dependent on $|V|, EO(f)$ but independent of $M(f)$.

As mentioned before, two common exact solutions for Problem 4 are linear programming or combining binary search with submodular minimization. However, neither of those two algorithms are strongly polynomial. In particular, there is no strongly-polynomial time solution for linear programming. Meanwhile, given a problem

**Algorithm 1** Density Improvement Framework for Problem 4

---

**Input:** A normalized supermodular function $f : 2^V \to \mathbb{R}$ via oracle
**Output:** $S^*$ maximizing $f(S)/|S|$.
  1: $S^0 \leftarrow V, t \leftarrow 0$
  2: **repeat**
  3:     $t \leftarrow t + 1, \beta^t \leftarrow d(S^{t-1})$
  4:     $S^t = \text{argmin}_{S \subset V} \beta^t |S| - f(S)$
  5: **until** $\beta^t |S^t| - f(S^t) = 0$
  6: **return** $S^{t-1}$

---

instance of Problem 4, one can binary search the optimum and answer the decision problem that given a parameter $\beta$, decide whether there exists one set $S$ with $f(S)/|S| > \beta$. However, the range to perform binary search and the termination condition both depend on $M(f)$. For example, for the simplest case that $f$ is nonnegative and integral, we have $M(f) = f(V) = \max_{S \subset V} f(S)$. Thus the optimum falls into the range $[0, f(V)]$ and for any two $S, T \subset V$ with $f(S)/|S|$ and $f(T)/|T|$ different, the minimum gap between $f(S)/|S|$ and $f(T)/|T|$ is $1/|V|^2$. This means the binary search takes $O(\log(M(f)|V|^2))$ iterations. Hence it has a dependence on $M(f)$ and is not strongly polynomial.[2]

Inspired by Dinklebach's algorithm [16], we give a simple strongly polynomial algorithm framework for a general normalized supermodular function $f$ in Algorithm 1. Each iteration minimizes a submodular function in strongly polynomial time [37]. And thus, we call it a density improvement framework as in each iteration the answer gets improved. More importantly, we will also show in numerical experiments that this in fact can be much faster than alternatives based on binary search commonly used in the literature.

The following standard result shows optimality at termination, of which proof is included in Appendix A.1.

LEMMA 2. *For a normalized, supermodular function $f : 2^V \to \mathbb{R}$, and a given parameter $\beta$, $\min_{S \subset V} \beta|S| - f(S) < 0$ if and only if there exists a set $S$ such that $f(S)/|S| > \beta$. As a result, $\min_{S \subset V} \beta|S| - f(S) = 0$ if and only if $\max_{S \subset V} f(S)/|S| \leqslant \beta$.*

Suppose Algorithm 1 runs for $T$ iterations. By Lemma 2, $\beta^T |S^T| - f(S^T) = 0$ suggests that

$$\max_{S \subset V} f(S)/|S| \leqslant \beta^T = f(S^{T-1})/|S^{T-1}|,$$

which means $S^{T-1}$ is optimal. We make one important observation that the size of $S^t$ is strictly decreasing. This is intuitive since we can view $\beta|S|$ as an $\ell_1$-norm penalty on $|S|$, thus with penalty coefficient $\beta$ increasing, the size of the solution to $\min_{S \subset V} \beta|S| - f(S)$ tends to decrease. By our algorithm design, $\forall t < T$,

$$\beta^t |S^t| < f(S^t), \tag{1}$$

because we terminate the algorithm once at a point we get $\beta^t |S^t| = f(S^t)$. As $f$ is normalized, for $t < T$, $S^t$ is non-empty. Hence Equation (1) implies $\forall t < T$,

$$\beta^t < f(S^t)/|S^t| = \beta^{t+1}, \tag{2}$$

where the equality follows from the definition of $\beta^t$. This means the sequence of $\beta$ is strictly increasing. By our algorithm design, for $\forall t > 0, S^t$ is the minimizer of $\beta^t|S| - f(S)$, hence we have $\forall t > 0$

$$\beta^t |S^t| - f(S^t) = \min_{S \subset V} \beta^t |S| - f(S) \leqslant \beta^t |S^{t+1}| - f(S^{t+1}). \tag{3}$$

Observe that via Equation (2), we have $\forall t < T - 1$,

$$f(S^t)/|S^t| = \beta^{t+1} < \beta^{t+2} = f(S^{t+1})/|S^{t+1}|.$$

which further shows that $\forall t < T - 1$,

$$\beta^{t+1}|S^{t+1}| - f(S^{t+1}) < 0 = \beta^{t+1}|S^t| - f(S^t). \tag{4}$$

Combine Equation (3) and (4), we get $\forall 1 \leqslant t < T - 1$

$$\beta^t(|S^t| - |S^{t+1}|) \leqslant f(S^t) - f(S^{t+1}) < \beta^{t+1}(|S^t| - |S^{t+1}|).$$

As $\beta^t < \beta^{t+1}$ for $\forall t < T$, we get for all $1 \leqslant t < T - 1$,

$$|S^t| > |S^{t+1}|.$$

Notice that if $T > 1$, then $d(S^1) = \beta^2 > \beta^1 = d(S^0) = d(V)$ where the last equality is because of our choice of $S^0$. Thus $S^1 \neq V$ and $|S^1| < |V| = |S^0|$.

Based on the discussion above, we have the following result.

THEOREM 2. *Assume the density improvement procedure terminates after $T$ iterations, then we have*

- $\forall t < T, \beta^t < \beta^{t+1}$.
- $\forall t < T - 1, |S^t| > |S^{t+1}|$.

*As a result, this procedure will terminate after at most $|S^0|+1 = |V|+1$ iterations and the algorithm runs in strongly polynomial time.*

This conclusion shows that Algorithm 1 will iteratively decrease the size of the solution. The supermodularity of $f$ ensures that Line 4 of Algorithm 1 can be done in strongly polynomial time. As a result, the whole procedure is strongly polynomial. We can see that in each iteration, when minimizing $\beta^t |S| - f(S)$, the minimization algorithm does not matter much as long as it is strongly polynomial.

Also here for simplicity of analysis, we take $S^0 = V$, but we can always start from some better initial sets and there is some potential to reuse information from previous solutions, which is sometimes more useful than solving the whole problem from scratch. The bound on the number of iterations is also rather loose, in other words, we believe in practice the number of iterations may be $o(|V|)$. Moreover, the supermodularity of $f$ is not necessary as long as $f$ has some special properties which enable a strong polynomial algorithm for minimizing $\beta|S| - f(S)$.

## 5 ANCHORED DENSEST SUBHYPERGRAPH

We now turn to concrete special cases of Problem 4 that focus on returning a dense subhypergraphs that are localized around a given seed set in a hypergraph.

PROBLEM 5 (ANCHORED DENSEST SUBHYPERGRAPH (ADSH)). *Given a hypergraph $\mathcal{H} = (V, \mathcal{E})$, a locality parameter $\varepsilon \geqslant 0$ and a seed set $R \subset V$, find a vertex set $S$ maximizing $d(S) = (e[S] - \varepsilon Vol(S \cap \bar{R})/2)/|S|$.*

PROBLEM 6 (ANCHORED DENSEST SUBHYPERGRAPH WITH FRACTIONAL VOLUME (ADSH-F)). *Given a hypergraph $\mathcal{H} = (V, \mathcal{E})$, a locality parameter $\varepsilon \geqslant 0$ and a seed set $R \subset V$, find a vertex set $S$ maximizing $\bar{d}(S) = (e[S] - \varepsilon \overline{Vol}(S \cap \bar{R}))/|S|$.*

---

[2]This statement holds for a general function $f$. For specific $f$, we may have that $M(f)$ is a simple function of $|V|$ or $|E|$ and binary search would be strongly polynomial.

These problems are inspired by ADS (Problem 3) and generalize it by applying to hypergraphs and including a locality parameter. They use the two different types of hypergraph volume and ADSH-F avoids situations where large hyperedges incur extremely large penalties. We focus results on ADSH in the interest of space.

## 5.1 A Flow-Based Exact Algorithm

We first introduce a flow-based exact algorithm that applies to the following problem that generalizes Problems 5 and 6:

$$\max \left( e[S] - p(S) \right) / |S|, \quad p : V \to \mathbb{R}_{\geqslant 0}. \quad (5)$$

We show how to solve this by reducing it to a sequence of generalized hypergraph $s$-$t$ cut problems, which can be solved in turn via reduction to graph $s$-$t$ cut problems using existing techniques [42].

Consider the decision version of Eq. (5). For a parameter $\beta$, there exists an $S$ such that $\left( e[S] - p(S) \right) / |S| > \beta$ if and only if there exists an $S$ such that $p(S) + \beta|S| - e[S] < 0$. We have that $e[S] = \overline{\text{Vol}}(S) - \sum_{e \in \mathcal{E}} g_e(S)$ where $g_e(S) = \min \frac{1}{|e|} \{|e \cap S|, \infty|e \setminus S|\}$, as

$$e[S] = \sum_{e \in \mathcal{E}} 1_{\{e \subset S\}} = \sum_{e \in \mathcal{E}} \left( \frac{|e \cap S|}{|e|} - g_e(S) \right) \quad (6)$$

$$= \sum_{v \in S} \sum_{e \ni v} \frac{1}{|e|} - \sum_{e \in \mathcal{E}} g_e(S) = \overline{\text{Vol}}(S) - \sum_{e \in \mathcal{E}} g_e(S).$$

We can therefore verify whether there exists an $S$ such that $p(S) + \beta|S| - e[S] < 0$ by solving a hypergraph min $s$-$t$ cut problem on the extended hypergraph $\mathcal{H}_\beta$ constructed as follows:

- Keep all of $\mathcal{H} = (V, \mathcal{E})$ and for each hyperegdge $e$, assign one splitting function $g_e(S) = \min \frac{1}{|e|} \{|e \cap S|, \infty|e \setminus S|\}$.
- Introduce one super source $s$ and create one edge $\{s, v\}$ with weight $\overline{\deg}(v)$ for each $v \in V$.
- Introduce one super sink $t$ and create one edge $\{v, t\}$ with weight $\beta + p(v)$ for each $v \in V$.

We focus here on the case where $\beta \geqslant 0$ since the optimal solutions to Problems 5 and 6 are always nonnegative. Note however that we can also handle $\beta < 0$ using slight adjustments to the construction above. We refer to edges directed connected to $s$ or $t$ terminal edges, denoted by $\mathcal{E}^{st}$. Their splitting function is the same as the cut function for a standard graph: the penalty is 0 if the edge is not cut and otherwise is equal to the weight of the edge. Every $S \subset V$ induces a hypergraph $s$-$t$ cut on $\mathcal{H}_\beta$ with value

$$\textbf{cut}(S \cup \{s\}) = \sum_{v \in \bar{S}} \overline{\deg}(v) + \sum_{v \in S} \left( p(v) + \beta \right) + \sum_{e \in \mathcal{E}} g_e(S) \quad (7)$$

$$= \overline{\text{Vol}}(\bar{S}) + \beta|S| + p(S) + \sum_{e \in \mathcal{E}} g_e(S)$$

$$= \overline{\text{Vol}}(\mathcal{H}) - e[S] + \beta|S| + p(S).$$

where the last equality is due to Eq. (6). We summarize as:

OBSERVATION 1. *The minimum $s$-$t$ cut of $\mathcal{H}_\beta$ is strictly smaller than $\overline{\text{Vol}}(\mathcal{H})$ if and only if there exists $S$ with $\left( e[S] - p(S) \right) / |S| > \beta$.*

For each $e \in \mathcal{E}$, the splitting function $g_e$ is submodular, cardinality-based and *asymmetric*. Under these conditions, previous work has shown how to reduce a generalized hypergraph $s$-$t$ cut problem to a graph $s$-$t$ cut problem. We include details here for completeness. For

this reduction, no change needs to be made to terminal edges, since by construction they already involve only two nodes. As shown in [25, 42], each $e \in \mathcal{E}$ can be replaced by the following *gadget*

- Introduce one auxiliary node $v_e$.
- For each $v \in e$, introduce a directed edge from $v$ to $v_e$ with weight $\frac{1}{|e|}$, and a directed edge from $v_e$ to $v$ with weight $\infty$.

This leads to a new directed graph $G_\mathcal{H}$ on an augmented node set. For any $S \subset V$, if we include $v_e$ on the same side of $S$, then we incur a directed cut penalty of $\infty|e \setminus S|/|e|$, otherwise the incurred directed cut penalty is $|e \cap S|/|e|$. The minimum $s$-$t$ cut solution in $G_\mathcal{H}$ will naturally place the auxiliary node $v_e$ in a way that incurs the minimum possible penalty subject to the placement of the original node set $V$. Therefore, for a node set $S \subseteq V$, the penalty incurred because of nodes in hyperedge $e$ is exactly $g_e(S)$.

With this core algorithmic step, what is left is to determine what $\beta$s to test. The density improvement framework introduced in Section 4 applies here and provides a strongly polynomial algorithm. One could also use binary search, as done in many densest subgraph variants. Observe that the answer falls in the interval $[-p(V), |\mathcal{E}|]$. When $p$ is integral or rational, there exists some predetermined smallest gap between any two possible non-equal values of $\left( e[S] - p(S) \right) / |S|$, and we can determine the termination condition accordingly. When $p$ is irrational, although this strategy fails, we can still apply parametric flow to solve it, as in [22].

**New results for DSG in vertex-weighted graphs.** We note in passing that our approach for solving Objective 5 implies more general results for solving densest subgraph problems in vertex-weighted graphs. The following problem was introduced in [22] and later considered in [17].

PROBLEM 7 (HEAVY AND DENSE SUBGRAPH PROBLEM (HDSP)). *Given an undirected graph $(G, V, E, w_V, w_E)$ without self-loops, where $w_V : V \to \mathbb{R}_{\geqslant 0}$ and $w_E : E \to \mathbb{R}_{\geqslant 0}$, find $S^* \subset V$ such that*

$$S^* = \operatorname*{argmax}_{S \subset V} \frac{e[S] + \sum_{v \in S} w_V(v)}{|S|},$$

*where $e[S]$ is the sum of the weights of edges fully contained in $S$.*

This problem explicitly considers weighted edges. Note that our approach for solving Objective 5 can easily be extended to weighted settings as well by scaling hyperedges (and the resulting edges in the reduced graph). While HDSP focuses only on standard graphs, our approach applies more generally to hypergraphs. Furthermore, while HDSP focuses on nonnegative vertex weights, our approach effectively deals with nonpositive vertex weights. Combining our techniques with Goldberg's flow network for HDSP [22] leads to the following stronger result.

OBSERVATION 2. *There is an efficient flow-based exact algorithm for any problem of the form $\max_{S \subset V} \left( e[S] + p(S) \right) / |S|$, where $p : V \to \mathbb{R}$ is a vertex function with no sign constraint.*

## 5.2 A Strongly-local Flow Algorithm

We now show how to design a *strongly-local* algorithm for Problem 5, meaning that the runtime depends only on the size of $R$. Showing how to obtain a runtime that is independent of global graph properties is the most technically challenging contribution of our paper. Our goal here is to strike a balance between obtaining

strong theoretical guarantees of this form while ensuring the algorithm is practical. Thus, rather than pursuing the tightest possible analysis, we focus on providing the simplest exposition that leads to a runtime that is bounded exclusively in terms of $|R|$ and $\text{Vol}(R)$.

We first provide high-level intuition as to why strongly-local algorithms are possible. For Problem 5, we have $p(S) = \varepsilon \deg(S \cap \bar{R})/2$. This means that in the directed graph $G_{\mathcal{H}}$ presented in Section 5.1, every vertex will have one directed edge from the source node $s$ with weight $\overline{\deg}(v)$ and one directed edge to the sink node $t$ with weight $\beta + \varepsilon \deg(v \cap \bar{R})/2$. When solving a maximum $s$-$t$ flow problem in this graph, if $\varepsilon$ is large enough we can pre-route a significant amount of flow and saturate many of the edges leaving the source node $s$. In particular, for large enough $\varepsilon$, pre-routing flow in this way will saturate all edges $(s, v)$ for each $v \in \bar{R}$. In the remaining residual graph, the $s$ will only be adjacent to nodes in $R$, and the total weight of edges leaving $s$ will be much smaller than the total weight of edges entering $t$. In this way, the maximum $s$-$t$ flow value will be bounded in terms of the size of $R$ (rather than the size of the whole graph), and by carefully solving a sequence of smaller flow problems "nearby" $R$ we will be able to find the minimum $s$-$t$ cut of the entire graph $G_{\mathcal{H}}$ without having to visit all of its nodes and edges. In what follows we provide complete details for formalizing this intuition. Formally, we will prove that when $\varepsilon \geqslant 1$, Problem 5 can be solved exactly by a strongly-local algorithm.

We assume throughout our analysis that $d(R) = \Omega(1)$. In other words, the subhypergraph induced by $R$ has a density lower bounded by some universal constant. This will simplify the technical exposition without significantly changing the analysis. We could alternatively weaken this to a natural assumption that $R$ contains at least one hyperedge, which would only change the analysis slightly.

As a warm-up we prove that when $\varepsilon \geqslant 2$, Problem 5 is equivalent to finding the densest subhypergraph within $R$. A strongly-local algorithm can then easily be obtained by considering only subsets of $R$. This provides additional intuition as to why strongly-local algorithms are possible for large enough $\varepsilon$.

LEMMA 3. *When $\varepsilon \geqslant 2$, $\max_{S \subset V} d(S) \Leftrightarrow \max_{S \subset R} d(S)$.*

The proof is provided in Appendix A.2. We now present a strongly-local algorithm for $1 \leqslant \varepsilon < 2$. We first bound the range of values containing the optimal solution $d^*$.

LEMMA 4. *Let $d^* = \max_{S \subset V} d(S)$, then for $1 \leqslant \varepsilon < 2$,*

$$\Omega(1) \leqslant \max_{S \subset R} d(S) \leqslant d^* \leqslant \bar{\Delta}(R).$$

The proof is provided in Appendix A.3. Hence, we only need to test those $\beta$s falling into this range of values that contains $d^*$. Recall that for a given parameter $\beta$, one can verify whether there exists one set $S$ such that $\left( e[S] - p(S) \right) / |S| > \beta$ by minimizing $p(S) + \beta|S| - e[S]$ and comparing the minimum to 0. Since $p(S) = \varepsilon \text{Vol}(S \cap \bar{R})/2$, we specifically minimize

$$\varepsilon \text{Vol}(S \cap \bar{R})/2 + \beta|S| - e[S]. \tag{8}$$

The following lemma bounds the size of the optimal set $S^* = \arg\max_{S \subseteq V} d(S)$, and the degree of nodes in $S^*$, in terms of quantities that depend only on $R$.

LEMMA 5. *When $\varepsilon \geqslant 1$, let $S^* = \arg\max_{S \subseteq V} d(S)$, then we have*

(1) $|S^*| \leqslant \overline{Vol}(R)$.

(2) $\forall v \in S^*, \deg(v) \leqslant O(\overline{Vol}(R) + \Delta(R))$.

Appendix A.4 provides a proof. This Lemma implies that when searching for the optimal $S^*$, we can ignore vertices with very high degrees. This is done by adding a directed edge from those vertices to $t$ with weight $\infty$. These edges will never be a part of the minimum $s$-$t$ cut, meaning that these vertices will never be part of $S$.

The main challenge is to find a minimum $s$-$t$ cut in $\mathcal{H}_\beta$ in a strongly-local manner. Recall from the construction of $\mathcal{H}_\beta$ in Section 5.1 that for every vertex $v \in V$, there is an edge $(s, v)$ and another edge $(v, t)$. This means that the minimum $s$-$t$ cut solution will have to cut one of these two edges for each vertex. Note that we can equivalently alter $\mathcal{H}_\beta$ so that each vertex in $\bar{R}$ has *either* an edge to the source $s$ or sink $t$ but not both. Concretely, for each $v \in \bar{R}$, we can remove the edge connected to $s$ with weight $\overline{\deg}(v)$ and decrease the weight of the edge to $t$ by $\overline{\deg}(v)$, so that the new weight is $\beta + \varepsilon \deg(v)/2 - \overline{\deg}(v)$. This is guaranteed to be nonnegative, since by Lemma 1 and the assumption that $\varepsilon \geqslant 1$ we have $\varepsilon \deg(v)/2 \geqslant \overline{\deg}(v)$. This adjustment will change the value of the minimum $s$-$t$ cut by $\text{Vol}(\bar{R})$, but will not change the minimizer. In what follows we assume we are working with this slightly altered hypergraph; we overload the notation and still call this $\mathcal{H}_\beta$.

Our strongly-local procedure works by starting with a subset of $\mathcal{H}_\beta$ and growing it as needed in search for a global minimum $s$-$t$ cut solution. We assume for this process that the hypergraph is given by oracle accesses. For each $v \in V$, let $\mathcal{N}_{\mathcal{E}}(v) = \{e \in \mathcal{E} : e \ni v\}$ be the set of hyperedges that $v$ belongs to and $\mathcal{N}_{\mathcal{E}}(S) = \bigcup_{v \in S} \mathcal{N}_{\mathcal{E}}(v)$. Let $V(e) = \{v : v \in e\}$ be the set of vertices that belongs to $e$ and $V(E) = \bigcup_{e \in E} V(e)$. Let $\mathcal{N}^{st}(v)$ denote those terminal edges incident to $v$ and $\mathcal{N}^{st}(S) = \bigcup_{v \in S} \mathcal{N}^{st}(v)$. Given a vertex $v \in V$ or a hyperedge $e \in \mathcal{E}$, we can efficiently query $\mathcal{N}_{\mathcal{H}}(v)$ or $V(e)$ respectively. Combining these two oracles, we can efficiently compute the vertex neighborhood of one vertex $\mathcal{N}_V(v) = \{u \in V : \exists e \text{ s.t. } u, v \in e\}$ and $\mathcal{N}_V(S) = \bigcup_{v \in S} \mathcal{N}_V(v)$. We also assume some simple metadata are pre-stored together with the hypergraph, for example we can query $\deg(v), \overline{\deg}(v)$ for any $v$, and $|e|$ for any $e$ in $O(1)$ time. Hence $\mathcal{N}^{st}(S)$ can be constructed efficiently.

Instead of building all of $\mathcal{H}_\beta$ explicitly and computing the minimum $s$-$t$ cut, we alternate between the following two steps:

- Compute a minimum $s$-$t$ cut $S_{\mathcal{L}}$ on a local hypergraph $\mathcal{L} \subseteq \mathcal{H}_\beta$.
- Expand the local hypergraph $\mathcal{L}$ based on the min $s$-$t$ cut $S_{\mathcal{L}}$ obtained in the above step.

This procedure ends at a point where we can certify that the $s$-$t$ cut on $\mathcal{L}$ is also a solution to the $s$-$t$ cut on the entire hypergraph $\mathcal{H}_\beta$. Concretely, let $\mathcal{L} = (V_{\mathcal{L}} \cup \{s, t\}, \mathcal{E}_{\mathcal{L}} \cup \mathcal{E}_{\mathcal{L}}^{st}, g)$ where $V_{\mathcal{L}} \subset V$ is a subset of the vertices of the hypergraph $\tilde{\mathcal{H}}$, $\mathcal{E}_{\mathcal{L}}$ is a subset of the hyperedges in $\mathcal{H}_\beta$ and $\mathcal{E}_{\mathcal{L}}^{st}$ is the set of the terminal edges in $\mathcal{H}_\beta$ between $V_{\mathcal{L}}$ and $\{s, t\}$, and $g$ is the set of splitting functions corresponding to $\mathcal{E}_{\mathcal{L}} \cup \mathcal{E}_{\mathcal{L}}^{st}$. We initialize $V_{\mathcal{L}}$ to be $R \cup \mathcal{N}_V(R)$, in other words, the seed set union its vertex neighborhood. We initialize $\mathcal{E}_{\mathcal{L}}$ to be $\mathcal{N}_{\mathcal{E}}(R)$, in other words, those hyperedges touching the seed set $R$. Finally we initialize $\mathcal{E}_{\mathcal{L}}^{st}$ to be the terminal edges connected to $V_{\mathcal{L}}$. When we grow $\mathcal{L}$, we always guarantee it remains a subhypergraph of $\mathcal{H}_\beta$, which means we always have

$$\textbf{min-st-cut}_{\mathcal{L}} \leqslant \textbf{min-st-cut}_{\mathcal{H}_\beta}. \tag{9}$$

---

**Algorithm 2** Strongly local method for solving Prob. 5 when $\varepsilon \geqslant 1$.

**Input:** $R, \varepsilon, \beta$, oracle access to a hypergraph $\mathcal{H}$.
**Output:** $S$ minimizing Objective (8) for $p(S) = \varepsilon \mathrm{Vol}(S \cap \bar{R})/2$.

1: $V_{\mathcal{L}} \leftarrow R \cup \mathcal{N}_V(R), \mathcal{E}_L \leftarrow \mathcal{N}_{\mathcal{E}}(R), \mathcal{E}_{\mathcal{L}}^{st} \leftarrow \mathcal{N}^{st}(V_{\mathcal{L}}), X \leftarrow R$
2: **repeat**
3: $\quad S_{\mathcal{L}}$ = Solve min-st-cut in $\mathcal{L}$ *Step 1*
4: $\quad S_{\mathrm{new}} = S_{\mathcal{L}} \setminus \big(X \cup \{s, t\}\big)$ *Step 2: Grow local hypergraph $\mathcal{L}$*
5: $\quad V_{\mathcal{L}} \leftarrow V_{\mathcal{L}} \cup \mathcal{N}_V(S_{\mathrm{new}})$
6: $\quad \mathcal{E}_{\mathcal{L}} \leftarrow \mathcal{E}_{\mathcal{L}} \cup \mathcal{N}_{\mathcal{E}}(S_{\mathrm{new}}), \mathcal{E}_{\mathcal{L}}^{st} \leftarrow \mathcal{E}_{\mathcal{L}}^{st} \cup \mathcal{N}^{st}(S_{\mathrm{new}})$
7: $\quad X \leftarrow X \cup S_{\mathrm{new}}$
8: **until** $S_{\mathrm{new}} = \emptyset$
9: **return** $S_{\mathcal{L}}$

---

By carefully choosing how the local hypergraph $\mathcal{L}$ grows, we can guarantee that the inequality will reach equality, without ever having to explore the entire hypergraph. This growing process expands $\mathcal{L}$ by considering nodes in $S_{\mathcal{L}}$ and adding all of its neighboring edges and nodes from $\mathcal{H}_{\beta}$ that are not already in the local hypergraph $\mathcal{L}$. We specifically have the following two update rules:

- Update the vertex set by setting $V_{\mathcal{L}} \leftarrow V_{\mathcal{L}} \cup \mathcal{N}_V(S_{\mathcal{L}})$.
- Update the edge set by setting $\mathcal{E}_{\mathcal{L}} \leftarrow \mathcal{E}_{\mathcal{L}} \cup \mathcal{N}_{\mathcal{E}}(S_{\mathcal{L}}), \mathcal{E}_{\mathcal{L}}^{st} \leftarrow \mathcal{E}_{\mathcal{L}}^{st} \cup \mathcal{N}^{st}(S_{\mathcal{L}})$.

To avoid adding the neighbor of one vertex multiple times, we keep a list $X$ and mark those vertices as *explored*. The algorithm ends when $S_{\mathcal{L}}$ does not introduce new vertices and edges. The whole procedure is summarized in Algorithm 2. Theorem 3 guarantees this will find the optimal minimum $s$-$t$ cut set, and Theorem 4 guarantees it will have a strongly-local runtime. We defer the proof for these results to the Appendix A.5 and A.6.

**THEOREM 3.** *When $\varepsilon \geqslant 1$, the optimal set $S$ returned by Algorithm 2 minimizes the objective* (8).

**THEOREM 4.** *For $\varepsilon \geqslant 1$, the local hypergraph $\mathcal{L}$ will contain $O((\overline{\mathrm{Vol}}(R) + |R|)\delta)$ hyperedges and at most $O((\overline{\mathrm{Vol}}(R) + |R|)\delta r)$ vertices where $\delta = O(\overline{\mathrm{Vol}}(R) + \Delta(R))$.*

## 6 EXPERIMENTS

We implement our proposed algorithms in Julia. Specifically, we solve the min. $s$-$t$ cut using a highest-label push-relabel algorithm with optimizations from [12]. We preprocess all the hypergraphs we use to remove dangling nodes, self-loops and multihyperedges.

To demonstrate the advantages and differences of the anchored densest subhypergraphs found by Problem 5 and 6, we compare them against running the anchored densest subgraph algorithm on the clique expansions of hypergraphs [15]. The specific clique expansions we consider are unweighted clique expansion (UCE) and weighted clique expansion (WCE). For WCE, each hyperedge $e$ will be replaced by one clique where each edge has weight $\frac{1}{|e|}$. For UCE, it is replaced by one clique where each edge has weight 1.

### 6.1 Density Improvement vs. Binary Search

To demonstrate that our Density Improvement Framework shown in Algorithm 1 has good performance in practice, we perform comparison experiments against binary search on five different

**Table 2: Comparison between our Density Improvement (DI) Framework shown in Algorithm 1 and the standard binary search (BS). Time is the running time in seconds and iterations represent the number of subproblems solved.**

| Datasets | $n$ | $m$ | $\overline{|e|}$ | DI | | BS | |
|---|---|---|---|---|---|---|---|
| | | | | time | iters | time | iters |
| Walmart | 87k | 65k | 6.9 | 6.4 | 9 | 18.5 | 43 |
| Trivago | 173k | 220k | 3.2 | 10.1 | 10 | 26.3 | 42 |
| Math SX | 153k | 563k | 2.6 | 19.5 | 8 | 89.5 | 47 |
| Ask Ubuntu | 82k | 114k | 2.3 | 2.6 | 10 | 8.7 | 43 |
| Amazon | 4.2M | 2.3M | 17.2 | 2239 | 10 | 9333 | 54 |

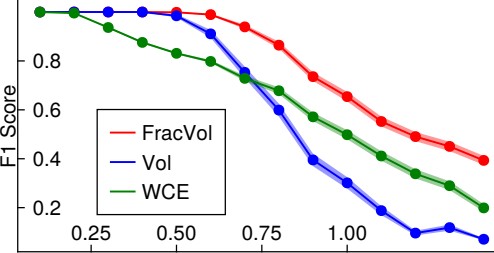

**Figure 1: Solving Probs 5, 6 (Vol, FracVol) outperforms Anchored Densest Subgraph (WCE) in planted set models. The x-axis represents difficulty $(m_1/m_2)$. Lines show mean F1 scores and bands show standard errors.**

real-world hypergraph datasets, Walmart Trips [2], Amazon Reviews [35], Trivago Clicks [14], Threads Ask Ubuntu and Threads Math SX [7]. Since the search range and termination condition of binary search gets complicated when the complexity of the objective function increases, here we simply study the densest subhypergraph problem. In other words, $f(S) = e[S]$. Each subproblem is solved by the same max. $s$-$t$ flow solver. We compare two methods' running time and number of subproblems solved.

The results are summarized in Table 2. Concretely, as is standard, for binary search, we let the search range be $[0, |\mathcal{E}|]$ and termination condition is when the search range becomes shorter than $1/(|V|(|V|-1))$ [28]. For our density improvement, we let $S^0 = V$. We can see that on all five datasets, density improvement shows about 3.5x speed up, which demonstrates it is practical.

### 6.2 Experiments with Planted Dense Sets

We first study the capacity of our objectives to find dense subhypergraphs on problems with planted dense subsets. Specifically, we build a graph with 1000 vertices and assign each vertex into one of the 30 clusters uniformly at random similar to a stochastic block model. Then we generate two kinds of hyperedges, $m_1$ hyperedges between clusters and $m_2$ hyperedges inside clusters. This allows us to plant 30 dense sets into this hypergraph. This is similar to scenarios for planted partitions in *uniform* hypergraphs where each hyperedge has the same size [20] and inspired by various ideas in random hypergraph and graph generation [1, 13, 23].

We let $m_2 = 50000$, and then compare our objectives with the baselines to see how well they can detect the underlying planted densest subhypergraphs when we vary $m_1$. The average hyperedge



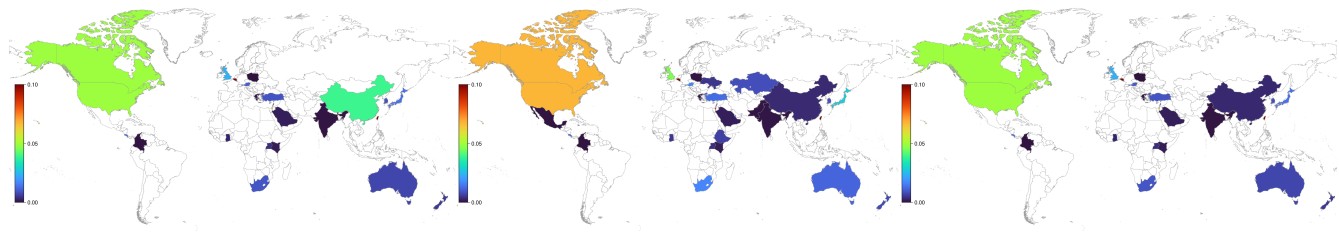

**(a) A 1543 vertex, density 22.34 subhypergraph from Chinese universities**

**(b) A 1923 vertex, density 19.76 subhypergraph from on UK universities**

**(c) A 1356 vertex, density 23.51 subhypergraph from the intersection**

**Figure 2: We show sets of domains as a colored map based on the number of domains associated with that region normalized by the total domains in the region. (Our attribution of domain to region is imperfect, but should capture general trends.)**

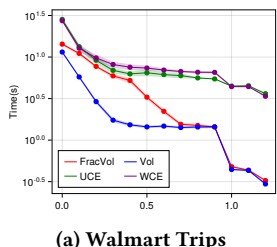

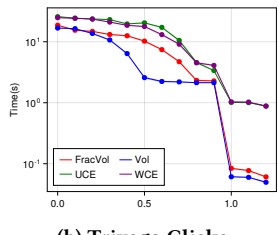

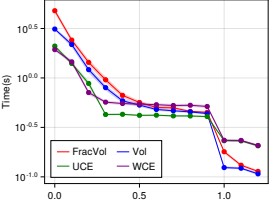

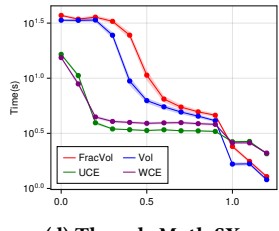

**(a) Walmart Trips**

**(b) Trivago Clicks**

**(c) Threads Ask Ubuntu**

**(d) Threads Math SX**

**Figure 3: Running time comparison. x-axis represents $\varepsilon$. We generate 100 different $R$ for each dataset and run each method on them. Here we report the mean and stderr of the running time. Specifically, each $R$ is generated by randomly sampling 10 seed nodes and then expand them to a set with 200 nodes using random walks.**

size in the hypergraphs we generate is 5.7. For each cluster, we generate 10 different seed sets $R$ by sampling 5% vertices from that cluster and performing length-2 random walks to grow it to a set with size equal to 1.5 times the cluster size. In total, we generate 300 seed sets. For each objective, we compute F1 score between the detected subhypergraph and the ground truth planted cluster. Appendix D.1 gives details for generating hyperedges.

The results are summarized in Figure 1. Here we do not show the result for UCE as it exhibits similar behavior with WCE empirically. We can see that when the planted dense structures are relatively clear, i.e. the ratio $\frac{m_1}{m_2}$ is relatively small, both Vol and FracVol penalty are able to perfectly detect the planted dense cluster while WCE can not. As is expected, with $m_1$ increasing, it is harder for all methods to recover the planted dense cluster but FracVol has a clear advantage throughout.

### 6.3 Densely linked domains on the web

We perform a case study on the web graph to show the local dense subgraph tools we build are useful in network analysis. We take the host-level webgraph data from Common Crawl (https://commoncrawl.org/blog/host-and-domain-level-web-graphs-oct-nov-jan-2020-2021). It contains 490 million nodes and 2.6 billion directed edges between hosts. We build a domain-level hypergraph by forming one hyperedge for each host within a domain. The contents of the hyperedge are all the domains linked from that host. We focus on the subgraph induced by the domain names of educational and academic institutions. Concretely, we take all domains ⋆.edu, ⋆.ac.⋆ or ⋆.edu.⋆. This hypergraph has 147k vertices and 138k hyperedges, with average hyperedge size 11.3. One common phenomenon of densest subgraph like objectives is that on real-world graphs, they

usually do not have large densest subgraphs. On this hypergraph, the densest subhypergraph contains 105 nodes, 103 US domains and 2 UK domains (Oxford, Cambridge) with density 45.73.

Our tools allow us to go beyond this simple set. Here we take domains from the UK and mainland China as reference sets respectively and vary $\varepsilon$ from 0.0 to 1.5 to find large, and reasonably dense subhypergraph. We identify one anchored densest subhypergraph with size 1923 and density 19.76 from the UK, and one anchored densest subhypergraph with size 1543 and density 22.34 from mainland China. By intersecting those two sets, we can get a denser subhypergraph with 1356 nodes and density 23.51 that spans universities throughout the world. This is illustrated in Figure 2.

### 6.4 Running Time Comparison

We compare runtime on real-world datasets and summarize results in Figure 3. All methods run faster when $\varepsilon$ is large and their running time sharply decreases when $\varepsilon$ enters the strongly local regime. Also, in general our anchored densest subhypergraph solvers are faster than clique expansion alternative when the hypergraph has a large mean hyperedge size, e.g. Walmart Trips and Trivago Clicks.

## 7 CONCLUSION

We propose two *localized* densest subhypergraph objectives and demonstrate their utility through experiments. Along the way, we also prove several interesting results for the general densest subgraph discovery problem. Future directions are making the algorithms scale to hypergraphs with billions of nodes and edges and futher exploring the space of *localized* objectives for different scenarios.

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

## A PROOFS

### A.1 Proof of Lemma 2

Since $f$ is normalized, if $\beta|S| - f(S) < 0$, then $S \neq \emptyset$. Therefore any $S$ satisfying $\beta|S| - f(S) < 0$ has $f(S)/|S| > \beta$. Meanwhile any $S$ with $f(S)/|S| > \beta$ naturally has $\beta|S| - f(S) < 0$.

Further, notice that $\beta|\emptyset| - f(\emptyset) = 0$, hence $\min_{S \subset V} \beta|S| - f(S) \leqslant 0$ always holds. As $\min_{S \subset V} \beta|S| - f(S) < 0$ and $\max_{S \subseteq V} f(S)/|S| > \beta$ are equivalent, the complement of them are also equivalent.

### A.2 Proof of Lemma 3

By our assumption, $d(R)$ is positive. Hence the optimal $S$ maximizing $d(S)$ has to intersect $R$, otherwise

$$\frac{e[S] - \varepsilon \text{Vol}(S \cap \bar{R})/2}{|S|} = \frac{e[S] - \varepsilon \text{Vol}(S)/2}{|S|} \leqslant 0.$$

For an $S$ that intersects $R$, let $A = S \cap R$ and $B = S \cap \bar{R} = S \setminus A$. Then

$$\frac{e[S] - \varepsilon \text{Vol}(S \cap \bar{R})/2}{|S|} \leqslant \frac{e[S] - \text{Vol}(S \cap \bar{R})}{|S|} \leqslant \frac{e[S \cap R]}{|S|}$$
$$\leqslant \frac{e[S \cap R]}{|S \cap R|} = \frac{e[A] - \varepsilon \text{Vol}(A \cap \bar{R})}{|A|}.$$

For the second inequality we use the fact that any hyperedge fully contained in $S$ and intersecting $S \cap \bar{R}$ is counted at least once in $\text{Vol}(S \cap \bar{R})$. The last equality follows from the definition of $A$. This inequality shows that for any set $S$ intersecting $R$, removing vertices outside $R$ will not make the answer worse. Thus it is equivalent to maximizing $d(S)$ over $S \subset R$.

### A.3 Proof of Lemma 4

On the one hand, we know that $d^* \geqslant \max_{S \subset R} d(S) \geqslant d(R) = \Omega(1)$ by assumption. On the other hand, observe that

$$d(S) = \frac{e[S] - \varepsilon \text{Vol}(S \cap \bar{R})/2}{|S|} \leqslant \frac{\overline{\text{Vol}}(S) - \frac{1}{2}\text{Vol}(S \cap \bar{R})}{|S|}$$
$$\leqslant \frac{\overline{\text{Vol}}(S) - \overline{\text{Vol}}(S \cap \bar{R})}{|S|} \leqslant \frac{\overline{\text{Vol}}(S \cap R)}{|S \cap R|} \leqslant \bar{\Delta}(R).$$

The first inequality relies on Eq. (6) and the second inequality follows from Lemma 1.

### A.4 Proof of Lemma 5

By Lemma 4, we have when $1 \leqslant \varepsilon < 2$,

$$d^* = \frac{e[S^*] - \varepsilon \text{Vol}(S^* \cap \bar{R})}{|S^*|} = \Omega(1),$$

which means

$$e[S^*] - \varepsilon \text{Vol}(S^* \cap \bar{R})/2 \geqslant \Omega(|S^*|)$$
$$\Rightarrow \overline{\text{Vol}}(S^*) - \sum_{e \in \mathcal{E}} g_e(S^*) - \varepsilon \text{Vol}(S^* \cap \bar{R})/w \geqslant \Omega(|S^*|)$$
$$\Rightarrow \overline{\text{Vol}}(S^*) \geqslant \sum_{e \in \mathcal{E}} g_e(S^*) + \varepsilon \text{Vol}(S^* \cap \bar{R})/2 + \Omega(|S^*|)$$
$$\Rightarrow \overline{\text{Vol}}(S^* \cap R) = \overline{\text{Vol}}(S^*) - \overline{\text{Vol}}(S^* \cap \bar{R})$$
$$\geqslant \sum_{e \in \mathcal{E}} g_e(S^*) + \left(\varepsilon \text{Vol}(S^* \cap \bar{R})/2 - \overline{\text{Vol}}(S^* \cap \bar{R})\right) + \Omega(|S^*|)$$
$$\Rightarrow \overline{\text{Vol}}(R) = \overline{\text{Vol}}(S^* \cap R) + \overline{\text{Vol}}(\bar{S}^* \cap R)$$
$$\geqslant \sum_{e \in \mathcal{E}} g_e(S^*) + \left(\varepsilon \text{Vol}(S^* \cap \bar{R})/2 - \overline{\text{Vol}}(S^* \cap \bar{R})\right)$$
$$+ \overline{\text{Vol}}(\bar{S}^* \cap R) + \Omega(|S^*|). \quad (10)$$

Recall that $\forall e \in \mathcal{E}, g_e(S) = \min \frac{1}{|e|}\{|e \cap S|, \infty|e \setminus S|\} \geqslant 0$, thus $\sum_{e \in \mathcal{E}} g_e(S^*)$ is nonnegative. By Lemma 1, we have $\text{Vol}(S^* \cap \bar{R}) \geqslant 2\overline{\text{Vol}}(S^* \cap \bar{R})$, which implies $\varepsilon \text{Vol}(S^* \cap \bar{R})/2 - \overline{\text{Vol}}(S^* \cap \bar{R}) \geqslant 0$ when $\varepsilon \geqslant 1$. Hence all four terms of RHS of (10) are nonnegative. This means

$$\overline{\text{Vol}}(R) \geqslant \sum_{e \in \mathcal{E}} g_e(S^*), \quad (11)$$
$$\overline{\text{Vol}}(R) \geqslant \varepsilon \text{Vol}(S^* \cap \bar{R})/2 - \overline{\text{Vol}}(S^* \cap \bar{R}), \quad (12)$$
$$\overline{\text{Vol}}(R) \geqslant \Omega(|S^*|). \quad (13)$$

Hence Ineq. (13) concludes that $|S^*| \leqslant O(\overline{\text{Vol}}(R))$.

To bound the degree of $v \in S^*$, we need to group the degree of each vertex into two subgroups contributed by hyperedges with different sizes. Let $\mathcal{E}_1 = \{e \in \mathcal{E} : |e| = 2\}, \mathcal{E}_2 = \mathcal{E} \setminus \mathcal{E}_1$. Let $\deg_i(v) = \sum_{e \in \mathcal{E}_i} 1_{\{v \in e\}}, \forall i \in \{1, 2\}$. In other words, we count the degree contributed by size-2 hyperedges separately. Accordingly, let $\text{Vol}_i(S) = \sum_{v \in S} \deg_i(v), \mathcal{N}_i(S) = \{e \in \mathcal{E}_i : |e \cap S| > 0\}, \forall i \in \{1, 2\}$.

For any $v \in S^* \cap R$, its degree is bounded by $\Delta(R)$. For any $v \in S^* \cap \bar{R}$, we consider its $\deg_1(v)$ and $\deg_2(v)$ separately. Because of Ineq. (12) and $\varepsilon \geqslant 1$, we have

$$\overline{\text{Vol}}(R) \geqslant \varepsilon \text{Vol}(S^* \cap \bar{R})/2 - \overline{\text{Vol}}(S^* \cap \bar{R})$$
$$\geqslant \frac{1}{2}\text{Vol}(S^* \cap \bar{R}) - \overline{\text{Vol}}(S^* \cap \bar{R})$$
$$= \sum_{v \in S^* \cap \bar{R}} \sum_{e \in \mathcal{E}: e \ni v} (\frac{1}{2} - \frac{1}{|e|})$$
$$\geqslant \frac{1}{6}\text{Vol}_2(S^* \cap \bar{R}),$$

which implies that for any $v \in S^* \cap \bar{R}$, $\deg_2(v) \leqslant \text{Vol}_2(S^* \cap \bar{R}) \leqslant 6\overline{\text{Vol}}(R)$.

By Ineq. (11), we have

$$2\overline{\mathrm{Vol}}(R) \geqslant 2\sum_{e \in \mathcal{E}} g_e(S^*) \geqslant 2\sum_{e \in \mathcal{E}_1} g_e(S^*)$$

$$= \sum_{e \in \mathcal{E}_1} \min\{|e \cap S^*|, \infty|e \setminus S^*|\}$$

$$= \sum_{e \in \mathcal{E}_1} 1_{\{|e \cap S^*|=1\}} \geqslant |\mathcal{N}_1(S^*) \cap \bar{S}^*|$$

where the first equality is due to the definition of $g_e$ and the second equality is due to $\mathcal{E}_1$ only contains size-2 hyperedges. In summary, for any $v \in S^* \cap \bar{R}$,

$$\deg_1(v) = |\mathcal{N}_1(v)|$$
$$= |\mathcal{N}_1(v) \cap S^*| + |\mathcal{N}_1(v) \cap \bar{S}^*|$$
$$\leqslant |S^*| + |\mathcal{N}_1(S^*) \cap \bar{S}^*|$$
$$= O(\overline{\mathrm{Vol}}(R)).$$

Thus for any $v \in S^* \cap \bar{R}$, we have $\deg(v) = O(\overline{\mathrm{Vol}}(R))$. Combined with the case $v \in S^* \cap R$ that we have discussed, we get the conclusion that for any $v \in S^*$, $\deg(v) = O(\overline{\mathrm{Vol}}(R) + \Delta(R))$.

## A.5 Proof of Theorem 3

Let $\mathcal{L}$ be the final local hypergraph when Algorithm 2 stops, and $S$ be the corresponding algorithm output. By our algorithm design, $S$ is the solution of **min-st-cut**$_\mathcal{L}$ and all the hyperedges adjacent to $S$ are included in the local hypergraph $\mathcal{L}$ as all vertices in $S$ are *explored*. Hence we have

$$\textbf{min-st-cut}_\mathcal{L} = \textbf{cut}_\mathcal{L}(S) = \textbf{cut}_{\mathcal{H}_\beta}(S)$$
$$\geqslant \textbf{min-st-cut}_{\mathcal{H}_\beta} \geqslant \textbf{min-st-cut}_\mathcal{L},$$

where the first equality is due to the optimality of $S$ on $\mathcal{L}$, the second equality is because those hyperedges can be cut by $S$ has already all been included in $\mathcal{L}$, the first inequality is due to the fact one $s$-$t$ cut has value not less than the min. $s$-$t$ cut, and the last inequality is because of Eq. (9). This sandwich results shows that **min-st-cut**$_\mathcal{L} = $ **min-st-cut**$_{\mathcal{H}_\beta}$ and $S$ is also the global optimum.

## A.6 Proof of Theorem 4

One key difficulty of proving Theorem 4 is that the vertex set returned by Algorithm 2 is the solution of the decision problem (8) instead of the original optimization Problem 5. To better distinguish them, let $S_\beta$ be the minimizer of the problem in Eq. (8) with parameter $\beta$ given by the decision problem, and let $S^*$ be the maximizer of Problem 5, $d^* := d(S)$. Lemma 4 shows the range of values where $d^*$ lies, hence this is also the range of $\beta$ values we need to test, in other words, $\Omega(1) \leqslant \beta \leqslant \bar\Delta(R)$.

Recall that $\delta = \overline{\mathrm{Vol}}(R) + \Delta(R)$. The high-level idea of the proof is that we prove the following two claims

- The set of *explored* nodes $X$ has size $O(|R| + \overline{\mathrm{Vol}}(R))$.
- The maximum degree of vertices inside $X$ is $O(\delta)$.

By design, Algorithm 2 only adds hyperedges to the local hypergraph if they are in the neighborhood of some node in $X$. Thus, if we can proves the two claims above, this guarantees the number of hyperedges inside $\mathcal{L}$ is bounded by $|X|\Delta(X) = O(\overline{\mathrm{Vol}}(R)\delta)$. Since

the rank of the hypergraph is $r$, the number of nodes inside $\mathcal{L}$ is $O(\overline{\mathrm{Vol}}(R)\delta r)$.

**Additional Notation.** To prove these two claims, we introduce some additional algorithmic notation. Recall that for our strongly-local algorithm we are working with the version of $\mathcal{H}_\beta$ where node $v \in \bar{R}$ has no edge from the source, but has an edge $(v, t)$ of weight $\beta + \varepsilon \deg(v)/2 - \overline{\deg}(v) \geqslant 0$. We call this the terminal edge or $t$-edge of node $v$. Let $\mathcal{L}_i = (V_i \cup \{s, t\}, \mathcal{E}_i \cup \mathcal{E}_i^{st})$ be the local hypergraph on which we solve the minimum $s$-$t$ cut problem in the $i$-th iteration. Let $S_i$ be the minimum $s$-$t$ cut set in the $i$-th iteration. Recall from Section 5.2 that we find the minimum $s$-$t$ cut in the hypergraph by reducing it to a graph minimum $s$-$t$ cut problem by replacing each hyperedge with a certain graph cut gadget. Let $G_i$ represent the reduced graph for $\mathcal{L}_i$, and $S_i'$ be the minimum $s$-$t$ cut set in $G_i$. Recall that $G_i$ contains all the same nodes as $\mathcal{L}_i$ in addition to new auxiliary nodes. The construction of $G_i$ is designed so that $S_i = S_i' \cap V_i$. None of the auxiliary nodes have edges to $s$ or $t$. Let $N_i$ be the set nodes that are added to $X$ for the first time in iteration $i > 1$. In other words, $N_i$ is the set of nodes in $\bar{R}$ whose $t$-edge was cut for the first time in iteration $i > 1$. When Algorithm 2 terminates, we have $X = R \cup \bigcup_{i=1}^{T} N_i$.

**Implicit flow claim.** In practice, a graph minimum $s$-$t$ cut is typically computed by solving the dual maximum $s$-$t$ flow problem. By the min-cut/max-flow theorem, every edge in the minimum $s$-$t$ cut is saturated by the maximum $s$-$t$ flow. The solution to these problems may not be unique, but this holds independent of which min-cut or max-flow we find. In practice we can use any minimum $s$-$t$ cut or maximum $s$-$t$ flow. However, we will prove the following result regarding the existence of a maximum $s$-$t$ flow with a special property, in order to later bound the size of the set $X$.

CLAIM 1. *In iteration $j > 1$ of Algorithm 2, there exists some maximum $s$-$t$ flow that saturates the $t$-edge of every node in $X \cap \bar{R} = \bigcup_{i=1}^{j} N_i$.*

PROOF. We prove the claim by induction. $N_1$ is exactly the set of nodes in $\bar{R}$ whose $t$-edge is cut by the minimum $s$-$t$ cut of $G_1$. By the min-cut/max-flow theorem, *every* maximum $s$-$t$ flow will saturate the $t$-edges of $N_1$, so the result holds for the base case $i = 1$.

For the induction hypothesis we assume that in iteration $j - 1$, there is some maximum $s$-$t$ flow in the local graph $\mathcal{L}_{j-1}$, call it $F_{j-1}$, that saturates the $t$-edge of every node in $\bigcup_{i=1}^{j-1} N_i$. Consider the next local graph $\mathcal{L}_j$. Its construction does not depend in any way on the specific flow function $F_{j-1}$. Rather, the way it which it expands $\mathcal{L}_{j-1}$ depends only on the set of nodes $N_{j-1}$ that were newly *explored* in the previous iteration. Our goal is to prove that there exists some maximum $s$-$t$ flow that saturates every node in $\bigcup_{i=1}^{j} N_i$. We do this by construction. Observe that $F_{j-1}$ is already a feasible flow for $\mathcal{L}_j$ since $\mathcal{L}_{j-1}$ is a subgraph of $\mathcal{L}_j$. Starting from $F_{j-1}$, we can search for augmenting paths along which to send additional flow through $\mathcal{L}_j$ until we eventually reach a maximum $s$-$t$ flow. We can assume without loss of generality that we never *unsaturate* a $t$-edge that was saturated by $F_{j-1}$, as there would never be any net gain to sending flow from $t$ back to another node.

Let $F_j$ denote the flow obtained by starting with $F_{j-1}$ and augmenting it until it is a maximum $s$-$t$ flow for $\mathcal{L}_j$. By the min-cut/max-flow theorem, it must saturate the $t$-edge of every node

in $N_j$, since nodes in $N_j$ are in the minimum $s$-$t$ cut set returned in iteration $j$. By construction, $F_j$ also saturates the $t$-edge of every node in $\bigcup_{i=1}^{j-1} N_i$. We therefore have the desired result that $F_j$ saturates the $t$-edge of every node in $\bigcup_{i=1}^{j} N_i$. □

**Bounding the size of $X$.** By the above claim, when Algorithm 2 terminates in iteration $T$, there exists a maximum $s$-$t$ flow that saturates the $t$-edge of every node in $X \cap \bar{R}$. Recall that the edges adjacent to $s$ have weights that sum up to $\overline{\mathrm{Vol}}(R)$, so the maximum flow through $\mathcal{L}_T$ is at most $\overline{\mathrm{Vol}}(R)$. The $t$-edge of every node in $X$ has weight at least $\beta = \Omega(1)$. Since every node in $X \cap \bar{R}$ has its $t$-edge saturated by *some* maximum $s$-$t$ flow, the number of nodes in $X \cap \bar{R}$ is at most $\frac{\overline{\mathrm{Vol}}(R)}{\beta} = O(\overline{\mathrm{Vol}}(R))$. Thus, $|X| \leqslant O(|R| + \overline{\mathrm{Vol}}(R)))$.

**Bounding the degree of vertices in $X$.** Lemma 5 shows that $\Delta(S^*) = O(\delta)$ where $\delta = \overline{\mathrm{Vol}}(R) + \Delta(R)$. As mentioned before, this means that when solving Problem 5, we can ignore those vertices with degree higher than this threshold. This implies that when solving the decision version, one can also tweak the decision problem a little bit and only focus on vertices with degree not higher than this threshold. In other words, $\Delta(S_\beta) \leqslant O(\delta)$. Formally, let $V_\delta$ be the set of vertices with degree $O(\delta)$, then by Lemma 5, we have

$$\max_{S \subset V} \frac{e[S] - \varepsilon \mathrm{Vol}(S \cap \bar{R})/2}{|S|} = \max_{S \subset V_\delta} \frac{e[S] - \varepsilon \mathrm{Vol}(S \cap \bar{R})/2}{|S|}.$$

So given a parameter $\bar{\beta}$, instead of solving the decision problem via computing

$$\min_{S \subset V} \beta|S| - e[S] + \varepsilon \mathrm{Vol}(S \cap \bar{R})/2,$$

we compute

$$\min_{S \subset V_\delta} \beta|S| - e[S] + \varepsilon \mathrm{Vol}(S \cap \bar{R})/2.$$

This can be simply achieved via keeping the same $s - t$ flow network construction and add one edge from those vertices in $V \setminus V_\delta$ to the super sink $t$ with weight $\infty$. In this way, we can guarantee all vertices inside $S_i$ have degree $O(\delta)$. As vertices inside $R$ has degree $O(\delta)$, $X$ has maximum degree as $O(\delta)$.

# B COUNTEREXAMPLE FOR GREEDY PEELING

We adopt the greedy peeling algorithm for Problem 2 mentioned in [11] (Theorem 3.1). For completeness, we restate it here. For a normalized nonnegative supermodular set function $f : 2^V \to \mathbb{R}_{\geqslant 0}$, we first initialize $S := V$ and then we recursively find $v = \arg\min_{v \in S} f(v \mid S - v)$ and update $S := S \setminus \{v\}$ until $S$ becomes empty. Here $f(v \mid S - v) := f(S) - f(S \setminus \{v\})$ is the marginal gain brought by element $v$ to the set $S$. We see that when $f(S) = e[S]$ as in the classical DSG, $f(v \mid S - v)$ becomes the degree of vertex $v$ in the subgraph $G[S]$.

Now we are ready to present one example which shows that greedy peeling may perform very poorly when $f$ is not guaranteed to be nonnegative. Here we give a counterexample on a graph, which is a special case of a hypergraph. Consider the graph in Figure 4, which contains two cliques linked by one edge. The clique on the left-hand side contains $a$ vertices and the clique on the right-hand side contains $b$ vertices. We let $b = 9a$. We denote the left

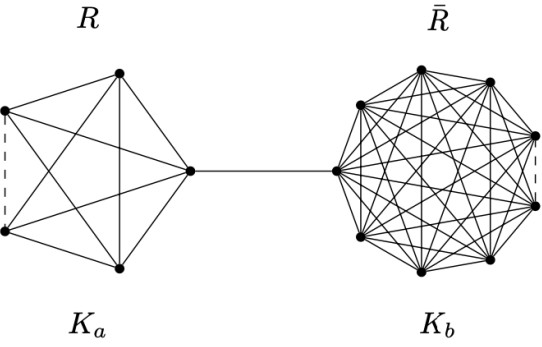

$$R \qquad \bar{R}$$

$$K_a \qquad K_b$$

**Figure 4: A counter example for greedy peeling.**

clique by $R$ and the right clique by $\bar{R}$ accordingly. We consider the following objective

$$\max_{S \subset V} \frac{e[S] - p(S)}{|S|}$$

where $p(v) = \frac{2}{3}b$ for $v \in \bar{R}$ and $p(v) = 0$ for $v \in R$. For this example, greedy peeling will first remove vertices from $R$ as the marginal gain brought by vertices from $\bar{R}$ is at least $\frac{1}{3}b = 3a$ and on the contrary the marginal gain brought by vertices from $R$ is at most $a$. Hence greedy peeling will not start peeling off vertices from $\bar{R}$ until the whole $R$ is peeled off. Then by symmetry, vertices from $\bar{R}$ will be peeled off in any random order.

We notice that when the intermediate subgraph only contains vertices inside $\bar{R}$, the objective is negative since $e[S] - p(S) = \binom{|S|}{2} - \frac{2}{3}b|S| < 0$.

When the intermediate subgraph still contains vertices from $R$, as pointed out before, at this time the whole clique inside $\bar{R}$ is also contained in the intermediate subgraph. Assume it contains $x$ vertices from $R$, then the objective now is

$$\frac{\binom{x}{2} + 1 + \binom{b}{2} - \frac{2}{3}b^2}{x + b} \leqslant \frac{\binom{a}{2} + 1 + \binom{b}{2} - \frac{2}{3}b^2}{b} < 0$$

as we let $b = 9a$. This means the peeling algorithm will only output negative answer on this example as we treat $0/0 = -\infty$. However the optimal solution is choosing $S = R$ and the optimum is $\frac{a-1}{2}$.

# C EXAMPLE FOR $\varepsilon < 1$

First we give some high-level intuition why strong locality may fail when $\varepsilon < 1$. Let us consider the degenerated case, graphs. Then the objective we focus on would be equivalent to maximizing

$$\frac{e[S] - \frac{\varepsilon}{2}\mathrm{Vol}(S \cap \bar{R})}{|S|},$$

which can be further transformed into

$$\frac{\frac{1}{2}\left(\mathrm{Vol}(S) - e(S, \bar{S})\right) - \frac{\varepsilon}{2}\mathrm{Vol}(S \cap \bar{R})}{|S|}$$

$$= \frac{\mathrm{Vol}(S \cap R)/2 - e(S, \bar{S})/2 + (1 - \varepsilon)\mathrm{Vol}(S \cap \bar{R})/2}{|S|}.$$

This means when $\varepsilon < 1$, including vertices outside $R$ can not only decrease the cut size but also add some volume reward.

$$\begin{array}{ccccc}
 & \overbrace{\phantom{R}}^{R} & & \overbrace{\phantom{\bar R \ \ \ \ \ \ \ \ \ \ \ }}^{\bar R} & \\
A = K_a & \underset{K_{a,b}}{\rule{2cm}{0.4pt}} & B = I_b & \underset{K_{b,c}}{\rule{2cm}{0.4pt}} & C = I_c \\
\text{deg} \qquad a+b-1 & & a+c & & b
\end{array}$$

**Figure 5: One counter example that shows strong locality is not guaranteed when $\varepsilon < 1$. It consists of three set of vertices $A, B, C$, and they contain $a, b, c$ vertices respectively. $A$ form a clique and $B, C$ are both independent sets. There is an edge between each vertex of $A$ and $B$, in other words there is a complete bipartite graph between them, the same holds for $B$ and $C$. The seed set $R$ is $A$. The degree for vertices in $A, B, C$ are $a - 1 + b, a + c$ and $b$ respectively.**

We show the following result which discourages strong locality when $\varepsilon < 1$.

THEOREM 5. *Assume the rank of $\mathcal{H}$, and $\varepsilon < 1$ are constants. There does not exist a universal polynomial $f(R)$ only with regard to quantities related to seed set $R$ such that the optimal $S^*$ of Problem 5 or 6 has size bounded by $f(R)$.*

PROOF. We give a counterexample on a normal graph, which is a special case of a hypergraph. Consider the example in Figure 5, where the seed set $R$ forms a clique and its density is $\frac{a-1}{2}$. Also, $\text{Vol}(R) = a(a - 1 + b)$ is independent of $c$. We take $b, c$ such that $c \gg b \gg a$.

Assume for a set $S$, $|S \cap A| = x, |S \cap B| = y, |S \cap C| = z$. Then the objective for $S$ becomes

$$\max_{\substack{a \geqslant x \geqslant 0, b \geqslant y \geqslant 0, \\ c \geqslant z \geqslant 0, x+y+z > 0}} \frac{\binom{x}{2} + y(x+z) - \frac{\varepsilon}{2}\left(y(a+c) + zb\right)}{x+y+z}. \quad (14)$$

We study when the maximum is achieved. Our proof strategy is to optimize over variables $z, y, x$ sequentially to eliminate them one by one. In other words, we find the optimal $z$ as a function of $y, x$, then the optimal $y$ as a function of $x$, and in the end the optimal $x$. We do not write out the functions explicitly but instead pay attention to the conditions when optimum is achieved.

Our proof heavily relies on the following observation.

OBSERVATION 3. *When we are maximizing any fraction with the form*

$$\frac{MX + N}{X + Q}$$

*over $X \in [L, U]$ where $M, N, L, U, Q$ are given real numbers, $L, U$ are nonnegative, and $Q$ is positive, the maximum is achieved on one of the two extreme points, $X = U$ or $X = L$.*

PROOF. We have

$$\max_{U \geqslant X \geqslant L} \frac{MX + N}{X + Q} = \max_{U \geqslant X \geqslant L} \frac{M(X + Q) + N - MQ}{X + Q}$$

$$= \max_{U \geqslant X \geqslant L} M + \frac{N - MQ}{X + Q}$$

$$= \max\left\{\frac{MU + N}{U + Q}, \frac{ML + N}{L + Q}\right\}. \quad (15)$$

$\square$

To avoid the corner case that $x+y+z = 0$, we divide the discussion into two subcases. First, we deal with the special case that $x = 0$. When $x = 0$, the objective (14) becomes

$$\max_{\substack{b \geqslant y \geqslant 0, c \geqslant z \geqslant 0, \\ y+z \geqslant 1}} \frac{yz - \varepsilon/2\left(y(a+c) + zb\right)}{y + z}. \quad (16)$$

Notice that if we plug in $y = b, z = c$ then we get $\frac{(1-\varepsilon)bc - \varepsilon ab/2}{b+c}$ which is positive as we assume $\varepsilon$ is a constant and $c \gg b \gg a$. This implies that any negative number cannot be the maximum of objective (16).

If $y = 0$, then we can see it is a constant function with value $-\frac{\varepsilon b}{2} < 0$. If $y > 0$, then objective (16) is equivalent to

$$\max_{b \geqslant y \geqslant 1} \max_{c \geqslant z \geqslant 0} \frac{yz - \varepsilon/2\left(y(a+c) + zb\right)}{y + z}. \quad (17)$$

Using Observation 3, we see that

$$\max_{c \geqslant z \geqslant 0} \frac{yz - \varepsilon/2\left(y(a+c) + zb\right)}{y + z}$$

$$= \max\left\{-\varepsilon(a+c)/2, \frac{yc - \varepsilon/2\left(y(a+c) + bc\right)}{y + c}\right\}.$$

As $-\varepsilon(a + c)/2$ is a negative constant, we have

$$(17) = \max_{b \geqslant y \geqslant 1} \frac{yc - \varepsilon/2\left(y(a+c) + bc\right)}{y + c}$$

$$= \max\left\{\frac{(1 - \varepsilon)bc - \varepsilon ab/2}{b + c}, \frac{c(1 - \varepsilon(1+b)/2) - \varepsilon a/2}{1 + c}\right\}$$

$$= \frac{(1 - \varepsilon)bc - \varepsilon ab/2}{b + c},$$

where in the second equality we use observation 3 again and the maximum is achieved when $y = b, c = z$. Now we turn to the main case that $x > 0$, and we can drop the constraint $x + y + z > 0$ and write objective (14) as

$$\max_{a \geqslant x \geqslant 1} \max_{b \geqslant y \geqslant 0} \max_{c \geqslant z \geqslant 0} \frac{\binom{x}{2} + xy + yz - \varepsilon/2\left(y(a+c) + zb\right)}{x + y + z}. \quad (18)$$

Apply observation 3 on variable $z$, we get

$$(18) = \max_{a \geqslant x \geqslant 1} \max_{b \geqslant y \geqslant 0} \max\left\{\frac{\binom{x}{2} + xy + yc - \varepsilon/2\left(y(a+c) + bc\right)}{x + y + c}, \right.$$

$$\left. \frac{\binom{x}{2} + xy - \varepsilon/2\left(y(a+c)\right)}{x + y}\right\}. \quad (19)$$

Again, we apply observation 3 on variable $y$, we get

$(18) = (19)$

$$= \max_{a \geqslant x \geqslant 1} \max\left\{ \frac{\binom{x}{2} + xb + bc - \varepsilon/2 \left(b(a+c) + bc\right)}{x + b + c}, \frac{\binom{x}{2} - \varepsilon bc/2}{x + c}, \right.$$

$$\left. \frac{\binom{x}{2} + xb - \varepsilon/2 \left(b(a+c)\right)}{x + b}, \frac{\binom{x}{2}}{x} \right\}.$$

Observe that by $x \leqslant a$, our assumption that $\varepsilon < 1$ is a constant and $c \gg b \gg a$, we have that $\frac{\binom{x}{2} - \varepsilon bc/2}{x+c}$ and $\frac{\binom{x}{2} + xb - \varepsilon/2(b(a+c))}{x+b}$ are both negative. Hence

$$(18) = \max_{a \geqslant x \geqslant 1} \max\left\{ \frac{\binom{x}{2} + xb + bc - \varepsilon/2 \left(b(a+c) + bc\right)}{x + b + c}, \frac{x-1}{2} \right\}.$$

When $x = a$,

$$\frac{\binom{x}{2} + xb + bc - \varepsilon/2 \left(b(a+c) + bc\right)}{x + b + c}$$

$$= \frac{\binom{a}{2} + (1 - \varepsilon/2)ab + (1 - \varepsilon)bc}{a + b + c}$$

$$\geqslant \frac{a-1}{2} = \max_{a \geqslant x \geqslant 1} \frac{x-1}{2}$$

as $c \gg b \gg a$. Therefore

$$(18) = \max_{a \geqslant x \geqslant 1} \frac{\binom{x}{2} + xb + bc - \varepsilon/2 \left(b(a+c) + bc\right)}{x + b + c}.$$

And whatever value $x$ takes, the maximum is achieved when $y = b, z = c$. Summarize all the cases above, the optimal $S$ has to contain the whole $B$ and $C$, which means $|S^*| \geqslant b+c$. Since $|R| = a$, $\mathrm{Vol}(R) = a(a - 1 + b)$, which are both independent of $c$, so we are unable to find a universal polynomial which can bound the size of optimal $S$.

## D   MORE IMPLEMENTATION DETAILS

### D.1   Hyperedge Generation for Experiments with Planted Dense Sets

Each hyperedge is generated in a similar way. Given a vertex pool $S$, we first sample two different vertices from $S$, and then we iteratively grow the hyperedge. In each iteration, with probability $p$ we stop the generating process and with probability $1 - p$ we sample another unique vertex from $S$ and continue to the next iteration. Once at a point the hyperedge reaches some pre-determined max size threshold, we also end the generating process. For those $m_1$ hyperedges between clusters, we let the vertex pool $S$ of the whole vertex set $V$ and for those $m_2$ hyperedges inside clusters, we pick a random cluster for each and set the vertex pool as vertices from that cluster. In this way, we plant 30 dense subhypergraphs in this 1000-vertex hypergraph. We let $m_2 = 50000, p = 0.2$ and set max hyperedge size as 12. With $m_1$ increasing, it will be much harder to detect the planted densest subhypergraphs as the background hypergraph gets denser and denser.

□

## E   ETHICS AND DATA

All of the data we use are publicly available and we do no mining of the data for specific human identifiable attributes. Some of the hypergraph data is based on public human activity, but we only use those experiments to calibrate performance on commonly used datasets. Our case study on the web graph is only based on linking patterns among web hosts and domains. Our dense subhypergraph tools have the potential to be used to identify extremal sets, which – like many general purpose mining tools – could be used maliciously to infer attributes that people would prefer stay secret if the information was represented by a dense graph. However, we believe that dense subgraph analysis and subhypergraph analysis is a common algorithmic framework that has substantial non malicious uses including novel studies of graph data and characterizing dense interconnections in biological networks.

