# OpenReview forum: "Densest Subhypergraph: Negative Supermodular Functions and Strongly Localized Methods"
_ACM.org/TheWebConf/2024/Conference — TheWebConf24 Oral_

### Official Review · Reviewer_4PXJ · 2023-11-19

**Novelty:** 5
**Technical Quality:** 5

**Review:**

**Quality and Clarity:**

- **Introduction**: The introduction effectively sets the stage for the importance of hypergraph analysis in the field. It would benefit from a more detailed discussion on the unique challenges that hypergraphs present, which this study aims to address.
- **Analysis**: The analytical portion is thorough and appears to be methodically conducted. Incorporating visual aids and illustrative examples would greatly assist in demystifying the complex concepts and enhancing the reader's comprehension.
- **Experiments**: The experimental section provides valuable insights but lacks a detailed description of the experimental setup and the datasets used. Expanding on these areas would help validate the robustness of the proposed methods.

**Originality**:

The paper introduces innovative approaches to hypergraph analysis, which is a relatively less explored area compared to standard graph analysis. The concept of localizing dense subgraphs using a tunable locality parameter is particularly novel and shows potential for new directions in research.

**Significance:**

The paper's contributions to hypergraph analysis are significant, particularly in their potential application to social network analysis and data clustering, as mentioned in the manuscript.

**Pros:**

- Advances the state-of-the-art in hypergraph analysis with new algorithms.
- Provides a rigorous theoretical foundation for the proposed methods.
- Demonstrates the applicability of the work through empirical evaluation.

**Cons:**

- Some parts of the paper might be too technical for readers not already familiar with hypergraph theory. A more thorough introduction to the foundational concepts could be beneficial. For example, the definitions of normal and fractional degrees and their related calculations of volume and maximum are presented in Table 1 with precise mathematical formalism, yet without supportive explanatory text or illustrative figures that could enhance understanding, particularly for novice readers. The addition of such explanatory aids could significantly lower the barrier to entry for the paper's broader audience.
    - how would the algorithm differentiate from techniques in [15] when the hypergraph graph reduces to the standard graph? Besides, what about their empirical performance when applying to standard graphs?
- The experiment section can be further enriched.
    - Although the paper presents experimental results on runtimes, there is a noticeable gap between the analysis provided and the paper's claims. The experimental section requires further refinement to better align with the assertions made about the strongly polynomial algorithm. For instance, while experiments demonstrate the efficiency of Algorithm 1, the discussion does not explicitly correlate these results to the theoretical underpinnings presented earlier in the paper. Clarification of how the empirical findings support Theorem 2 would greatly strengthen the paper's contribution and provide a more comprehensive validation of the proposed methods.
    - Further, how would the density improvement framework perform on standard graphs? Is it still useful in standard graphs, or just restricted to hypergraphs? More experiments could be helpful, e.g., comparing its runtime on standard graphs with SOTA algorithms, e.g., [1].
    - How does the density improvement framework perform (empirically) compared to LP (or convex programming) based solutions, e.g. [2].
- Sections 2 through 5 of the paper, while logically structured with pseudocode and formulaic calculations, lack concrete examples that could significantly enhance the reader's understanding of the algorithms and principles. The inclusion of step-by-step examples would be particularly beneficial. Such examples can illustrate the application of the algorithms in practical scenarios, providing a more tangible connection between the theoretical concepts and their practical utility. This approach would not only aid in demystifying the abstract parts of the paper but also make the methodologies more accessible to a wider audience.

[1] Xu, Yichen, et al. "Efficient and Effective Algorithms for Generalized Densest Subgraph Discovery." *Proceedings of the ACM on Management of Data* 1.2 (2023): 1-27.

[2] Danisch, Maximilien, T-H. Hubert Chan, and Mauro Sozio. "Large scale density-friendly graph decomposition via convex programming." *Proceedings of the 26th International Conference on World Wide Web*. 2017.

**update** After checking the author's response, I changed novelty from 4 to 5.

**Questions:**

Could you elaborate on the specific challenges that hypergraphs present in analysis, which are not encountered in standard graph analysis?

How do the density improvement techniques perform on standard graphs?

Would it be possible to include more visual aids or examples in the paper?

Can you provide further details on the experimental setup and the datasets used?

**Reviewer Confidence:**

3: The reviewer is confident but not certain that the evaluation is correct

**Scope:**

3: The work is somewhat relevant to the Web and to the track, and is of narrow interest to a sub-community

---

### Official Review · Reviewer_mApD · 2023-11-20

**Novelty:** 6
**Technical Quality:** 5

**Review:**

In this paper, the authors study dense subgraph discovery (DSD). DSD, aiming to find a dense component in a given graph, is a fundamental graph-mining primitive, which has a lot of real-world applications. The most well-studied optimization models for DSD would be the densest subgraph problem (DSP), where given an undirected graph G = (V,E), we are asked to find a vertex subset S that maximizes the so-called density (i.e., half the average degree of the induced subgraph). Unlike most of the other optimization models for DSD, DSP can be solved exactly in polynomial time. To date, a number of interesting generalizations of DSP have been introduced and addressed in the literature. The generalizations, on which the present paper is based, are the densest subhypergraph problem (DSHP) and the anchored densest subgraph problem (ADSP). The first one is a natural generalization of DSP, wherein the input is a hypergraph and the definition of the density is generalized, while in the second one, the input graph is associated with two vertex subsets called the anchored node set and the reference node set (the first is contained in the latter), and the goal is to find a densest subgraph that contains the anchored nodes but also close to the reference nodes. The first model is inspired by the capability of hypergraphs to represent more complex relationships, while the second model enables dense subgraph discovery around a specific subset of interest.

In this paper, the authors introduced common generalizations of the above two (strictly speaking, the latter is restricted a bit). The generalization, on which the authors mainly focus, is called the anchored densest subhypergraph (ADSH), where ADSP is naturally extended to hypergraphs but the anchored node set is always fixed to the emptyset. Therefore, the generalization seeks a densest subhypergraph that is close to the given reference nodes. The noteworthy feature of the generalization is the existence of the so-called locality parameter, which specifies how the output should be close to the given reference nodes. The authors first present a flow-based exact algorithm for ADSH, based on the flow-based exact algorithms for DSP and its variants and the recent development of computing a minimum s-t cut on hypergraphs. This result remains valid even for some generalizations of ADSH, containing DSP with node (positive and negative) profits. The main algorithmic contribution of this paper is to develop a strongly-local exact algorithm for ADSH, meaning that the algorithm outputs an optimal solution to the problem in time depending only on the parameters due to the reference node set. The algorithm runs in an iterative fashion: In the first iteration, the algorithm stores the subhypergraph consisting of the reference nodes and their neighbors, and the later iterations, it computes the minimum s-t cut in the current subhypergraph and expands the subhypergraph using the information of the cut computed just now. Also, in the preliminary discussion, the authors consider a quite generalization of ADSP, which they refer to as the densest supermodular subset with possible negative values. In the problem, given a normalized supermodular function f over a finite set V, we aim to find subset S of V that maximizes f(S)/|S|. This problem is also a generalization of the recently-introduced problem called the densest supermodular subset (DSS). DSS is invented as a generalization of DSP, and the supermodular function is assumed to be nonnegative and monotone. The above generalization removes the assumption, and hence it contains ADSH and some other existing problems as special cases. For this generalization, the contribution of the present paper is two fold. The first result is a simple reduction of the problem to DSS in terms of exact computation, meaning that we can solve even the above generalization through solving DSS. The second result is a strongly-polynomial exact algorithm for DSS. Although DSS is known to be polynomial-time solvable, there is no known strongly-polynomial exact algorithm. The authors designed it, based on Dinklebach's algorithm. Specifically, unlike the existing weakly-polynomial algorithms based on binary search over the objective values, the algorithm actively specifies the sequence of supermodular maximization problems to be solved.

Computational experiments using synthetic and real-world hypergraphs demonstrate the effectiveness of the proposed algorithms. The proposed strongly-polynomial exact algorithm for DSS is compared with the existing weakly-polynomial exact algorithm based on binary search, on a special case of DSS, i.e., DSHP. Note that for DSHP, even the weakly-polynomial algorithm becomes strongly polynomial, and therefore, there is no benefit of using the proposed algorithm in theory. However, the empirical results demonstrate that the proposed algorithm decreases the number of iterations needed and also the overall running time. The proposed strongly-local algorithm for ADSH is compared with the baseline methods that project the given hypergraph into an ordinary graph using clique expansions and solve ADSP exactly on the graph. The results on synthetic hypergraphs show that the proposed algorithm tends to detect a dense component more accurately, while the results on real-world hypergraphs verifies the scalability of the proposed method, particularly when the locality parameter is reasonably large. Finally, the authors conduct a case study using a web graph and suggest the possibility to use the proposed methodology in the web graph mining.

Strength:
- The paper addresses a fundamental graph-mining primitive (i.e., DSD). The generalizations of DSP introduced in this paper are all reasonable.
- The novelty of algorithmic contributions is sufficient. In particular, the strongly-local exact algorithm for ADSH and the strongly-polynomial exact algorithm for DSS are novel.
- The paper is technically sound. The reviewer does not come across any technical flaw.
- The paper is well-written and easy to follow. There are sufficient descriptions about how the present study contributes to the web graph mining.

Weakness and suggestions for improvements:
- The significance of the strongly-polynomial exact algorithm for DSS is rather limited, compared with that of the strongly-local exact algorithm for ADSH. In theory, such an algorithm is appreciated, but all the concrete examples of DSS have objective values polynomial in the input size. Therefore, the reviewer does not see the practical gain of the result. The proposed algorithm is effective when the objective value might be super-polynomial of the input size, but are there any examples?
- In the experimental evaluation, only the scalability is evaluated on real-world hypergraphs. Why don't the authors compare the accuracy of the algorithms? As the proposed strongly-local exact algorithm outputs an exact solution, it would always beat the baselines, but still it is worth presenting the concrete objective values. Currently the accuracy is compared using the ground-truth structure of synthetic hypergraphs. The performance of the proposed algorithm for ADSH is outperformed by the baseline, particularly when the instance is difficult. As the proposed algorithm for another version of ADSH performs even better, don't the authors consider presenting the results of the variant rather than those of ADSH (in the theory part)? The case study looks interesting and would be appreciated by the readers, the interpretation of the results is slightly ambiguous. In particular, the authors take the intersection of the subhypergraphs obtained, but ideally, should the final subhypergraph be obtained as the output for (one of) the reference node sets?
- In the theory part, there are some inaccurate or ambiguous descriptions. As mentioned above, ADSP is the problem associated with the anchored node set and the reference node set. The anchored nodes must be contained in the output, while the reference nodes have no such hard constraint. However, in the present paper, the special case where the anchored node set is empty is referred to as ADSP. Therefore, the output is no longer anchored, and the problem should be called like the referenced densest subgraph problem. Is there any obstacle to get the same result for the generalization of the original ADSP? This point should be explained. Another issue is found in the definition of g_e(S) in Line 482. If e \subset S, the second term of min becomes \infty times 0, which is undefined. Also from Line 672, the authors mention some additional (yet reasonable) assumptions to get the strongly-local algorithm for ADSH. These assumptions and the others should be put together at the beginning of the presentation of the algorithm.
- The writing can be still improved. Please see the list of minor comments below.
- Some references are missing. DSHP is introduced first by [HK95] rather than Hu et al. DSD is used for community detection tasks over web-based datasets also by [K+19]. Spectral methods and heat-kernel diffusions for the ordinary graphs are generalized also by [T+22] and [I+20]. Only the strongly-polynomial algorithm for submodular minimization by Orlin is mentioned, but the first algorithms are by [I+01] and [S00] and there exists a much faster algorithm [L+15].
[HK95] Huang and Kahng, When clusters meet partitions: New density-based methods for circuit decomposition, EDTC 1995.
[K+19] Kawase et al., Graph mining meets crowd sourcing: Extracting experts for answer aggregation, IJCAI 2019.
[T+22] Takai et al., Hypergraph clustering based on PageRank, KDD 2020.
[I+20] Ikeda et al., Finding Cheeger cuts in hypergraphs via heat equation, Theoretical Computer Science, 2022.
[I+01] Iwata et al., A combinatorial strongly polynomial algorithm for minimizing submodular functions, Journal of the ACM, 2001.
[S00] Schrijver, A combinatorial algorithm minimizing submodular functions in strongly polynomial time, Journal of Combinatorial Theory, Series B, 2000.
[L+15] Lee et al., A faster cutting plane method and its implications for combinatorial and convex optimization, FOCS 2015.

Minor comments:
- Mentioning a concrete value of the locality parameter (i.e., 1) in the abstract would not be helpful for the readers, as they do not know its scale.
- DSP with node weights (or some similar ways to call it) would remind the readers about the density with the node-weighted denominator.
- Line 120: It should be mentioned that the algorithm is proposed by Boob et al.
- Line 124: It should be mentioned that DSS is introduced by Chekuri et al.
- Line 172: Should e <= 2 be replaced by e = 2?
- Line 222: w_{uv} should be formally defined (i.e., w_{uv} = w({u,v})).
- Line 299: e(S,-S) -> cut_G(S).
- Line 354: \beta^t is confusing, as \beta usually represents a scalar. The reviewer would use \beta_t or \beta^(t).
- Line 417: Equation -> Equations
- Line 463: The second term should be divided by 2, as in Problem 5.
- Equation (5): The function p should satisfy some properties (e.g., p(S) = \sum_{v\in S}p(v)).
- Line 519: "cardinality-based" and "asymmetric" should be defined, and "previous work" should be specified.
- Line 548: Objective -> Problem
- Line 572: "vertex function" is ambiguous.
- Lines 586 and 590: deg -> Vol
- Line 588: every vertex -> every vertex from the original hypergraph
- Line 620: solution -> value
- Line 623: \Omega(1) <= max is not a standard notation.
- Line 633: Redundant space exists after argmax.
- Line 639: The inequality should be equality.
- Line 666: V(e) is redundant, as it equals e.
- Line 669: \mathcal{N}_\mathcal{H} is not defined.
- Line 675: Should "efficiently" be O(|S|)?
- Algorithm 2: The definition of S_\mathcal{L} is ambiguous, and S_\mathcal{L} in the definition of S_new should be \mathcal{N}_V(S_\mathcal{L}).
- The descriptions of Steps 1 and 2 in the algorithm look strange.
- Line 725: optimal minimum -> minimum
- Line 727: Appendix -> Appendices
- Line 742: Problem -> Problems
- Table 2: -|e| is not defined.

**Questions:**

Please address the following question in addition to the above ones.
- Is the proposed strongly-polynomial exact algorithm effective for the ordinary DSP? Of course, as mentioned above, the algorithm has no benefit in theory, but it is really appreciated by the community if the algorithm works better in practice than the usual binary-search based algorithm.

**Reviewer Confidence:**

4: The reviewer is certain that the evaluation is correct and very familiar with the relevant literature

**Scope:**

3: The work is somewhat relevant to the Web and to the track, and is of narrow interest to a sub-community

---

### Official Review · Reviewer_Cyde · 2023-11-24

**Novelty:** 5
**Technical Quality:** 5

**Review:**

Pros:
1. This paper studied two localized densest subhypergraph objectives and demonstrate their utility through experiments. A reduction that leads to efficient exact algorithms for a more general version of the densest supermodular subset (DSS) problem is given, which captures several dense subgraph problems that are not special cases of the standard nonnegative DSS problem. A strongly polynomial algorithm for DSS and a flow-based exact algorithm for finding the densest subset of a node-weighted graph or hypergraph are proposed. A parametric formulation of the discrete objective function underlying localized densest subgraph discovery is established.
2. The presentation of this manuscript is well-prepared.

Cons:
1.	In the related work section, the references of Problem 3 are missing.
2.	Several problems are mentioned in the paper, and I suggest to add graphs or examples to illustrate the key problems, while presenting the relationships between the various problems more intuitively.
3.	The correlation between the sections is not strong, and the structure of the paper needs to be optimized and adjusted.

**Questions:**

See above.

**Reviewer Confidence:**

2: The reviewer is willing to defend the evaluation, but it is likely that the reviewer did not understand parts of the paper

**Scope:**

3: The work is somewhat relevant to the Web and to the track, and is of narrow interest to a sub-community

---

### Official Review · Reviewer_UiMd · 2023-11-24

**Novelty:** 6
**Technical Quality:** 6

**Review:**

The paper studies versions of the classic densest subgraph problem. It has the following main result: It presents a local algorithm that takes a given set of seed nodes $S$ and finds the densest subhypergraph ``close'' to $S$; this algorithm runs in strongly local time (i.e., time polynomial in the size of the subgraph returned). Additionally, the paper also gives new global algorithms for vertex-weighted graphs and a reduction for maximizing the density of (possibly negative) supermodular functions.

The paper is on a highly interesting topic. The setting in which one is given an initial solution and the goal is to find a solution that is ``close'' to the initial one gained significant attraction recently, so the paper's topic is quite timely. Additionally, I think the setting with hypergraphs is nice and the results are non-trivial to obtain (even though they are also not exceptionally creative). The experimental evaluation is convincing and I like the case study. The paper is also very nicely written.

Overall, I think this is a very nice contribution to the WebConf and I would be happy to see the paper accepted.

*Comment after the rebuttal:* I have read the authors' rebuttal.

Comments for the authors:
- Problem 5 and 6: I would appreciate if a sentence was added saying that $\bar{R}$ is the complement of $R$. Since there are so many bars, I initially thought that it was a typo and was confused why the definition would make any sense.
- Line 217: There is a space missing before "(DSG)".
- Theorem 1: Perhaps one could rephrase the theorem statement slightly such that it becomes clearer what is meant by the "difference". The proof makes it clear, but initially I was confused because in the line above it is argued that the functions are the same.
- Also Theorem 1: I wonder whether the statement with the argmax f(S)/|S| = argmax g(S)/|S| is correct when there is more than one optimal solution (e.g., multiple disjoint subgraphs with the same densities)? Perhaps the theorem statement needs a little bit of rephrasing to take this case into account.

**Questions:**

I have no questions.

**Ethics Review Description:**

-

**Reviewer Confidence:**

4: The reviewer is certain that the evaluation is correct and very familiar with the relevant literature

**Scope:**

4: The work is relevant to the Web and to the track, and is of broad interest to the community

---

### Official Review · Reviewer_Svmx · 2023-11-26

**Novelty:** 5
**Technical Quality:** 4

**Review:**

This paper provides a variety of theoretical and algorithmic advancements to the problem of dense sub(hyper)graph discovery.


The main strengths of the paper are:

S1) Relevant advancements to the state of the art.

S2) Good novelty.

S3) Significant technical contribution.


The paper also comes with a couple of weaknesses (with the second one being particularly important):

W1) Presentation of the paper contributions can be improved. In particular, the paper provides several theoretical advancements. it would be nice to have them precisely, clearly and concisely summarized in some way (a table, perhaps).

W2) The experimental evaluation is somewhat weak as several theoretical results are not supported by it, namely: comparison with Charikar-like fast approximation algorithm for dense sub(hyper)graph discovery, handling (hyper)graphs with arbitrary node weights (along with a comparison to existing methods that handle nonnegative node weights only), handling the scenario of localized dense sub(hyper)graph discovery. The latter is particularly critical, as the scenario of localized dense sub(hyper)graph discovery constitutes a significant part of the paper.

**Questions:**

Please motivates W2) (if there are specific motivations of why those experiments have not been performed).

**Ethics Review Description:**

No ethical issues.

**Reviewer Confidence:**

2: The reviewer is willing to defend the evaluation, but it is likely that the reviewer did not understand parts of the paper

**Scope:**

3: The work is somewhat relevant to the Web and to the track, and is of narrow interest to a sub-community

---

### Decision · Program_Chairs · 2024-01-22

**Decision:**

Accept (Oral)

**Comment:**

The paper introduces the notion of anchored densest subgraph problem for hypergraphs. Then it provides a new algorithm for the local and the global densest subgraph for hypergraphs.

 The paper is on a relevant topic and provides several new theoretical insights on it.

 Overall, the reviewers found the paper nice and interesting and think that it would be a nice addition to the conference program.

 During the discussion the reviewers had some interesting suggestions to improve the quality of the paper:
 - add additional suggested related works
 - extend the experimental analysis to study empirically most of the presented algorithm